

# Nitrite Cycling in the Primary Nitrite Maxima of the Eastern Tropical North Pacific

Nicole M. Travis[1], Colette L. Kelly[1], Margaret R. Mulholland[2], Karen L. Casciotti[1]

[1] Earth System Science, Stanford University, Stanford CA, 94305, USA
[2] Department of Ocean, Earth and Atmospheric Science, Old Dominion University, Norfolk VA, 23529, USA

*Correspondence to:* Nicole M. Travis (ntravis@stanford.edu)

**Abstract.** The primary nitrite maximum (PNM) is a ubiquitous feature of the upper ocean, where nitrite accumulates in a sharp peak at the base of the euphotic zone. This feature is situated where many chemical and hydrographic properties have strong gradients and the activities of several microbial processes overlap. Near the PNM, four major microbial processes are active in nitrite cycling: ammonia oxidation, nitrite oxidation, nitrate reduction and nitrite uptake. The first two processes are mediated by the nitrifying archaeal/bacterial community, while the second two processes are conducted by phytoplankton. The overlapping spatial habitats and substrate requirements for these microbes have made understanding the formation and maintenance of the PNM difficult. In this work, we leverage high resolution nutrient and hydrographic data and direct rate measurements of the four microbial processes to assess the controls on the PNM in the Eastern Tropical Pacific. The depth of the nitrite maxima showed strong correlations with several water column features (e.g., top of the nitracline, top of the oxycline, depth of the chlorophyll maxima), whereas the concentration of the nitrite maxima correlated weakly with fewer water column features (e.g. nitrate concentration at the nitrite maximum). The balance between microbial rate processes active in nitrite cycling was a poor predictor of the concentration of the nitrite maximum, but rate measurements showed that nitrification was a major source of nitrite in the ETNP, while phytoplankton release occasionally accounted for large nitrite contributions near the coast. The temporal mismatch between rate measurements and nitrite standing stocks suggests that studies of the PNM across multiple time scales are necessary.

**Short Summary (500 char.)** The primary nitrite maximum is a ubiquitous upper ocean feature where nitrite accumulates, but we still do not understand its formation and the co-occurring microbial processes involved. Using correlative methods and rates measurements, we found strong spatial patterns between environmental conditions and depth of the nitrite maxima, but not the concentration of the maxima. Nitrification was a major producer of nitrite, with occasional high nitrite production from phytoplankton near the coast.

## 1 Introduction

Nitrogen (N) availability often controls ocean productivity through its role as a limiting nutrient. In marine systems, nitrate makes up over 88% of the bioavailable ('fixed') N pool, with dissolved organic N representing the next largest pool of fixed N (Gruber, 2008). However, the vertical distributions of these species render them unavailable to many



of the microbes that require them, e.g., nitrate is depleted in euphotic surface waters where primary production is
confined, but abundant in waters below the euphotic zone. Other inorganic fixed N species, e.g., nitrite and
ammonium, are present in smaller quantities, but their production and consumption are tightly coupled in the marine
environment. In the upper ocean, the nitracline demarcates a spatial transition where nitrate, nitrite and ammonium
may all be available to microbes simultaneously. In particular, the primary nitrite maximum (PNM) is a ubiquitous
feature of the upper ocean, where nitrite often accumulates to concentrations near 300 nM, although concentrations as
high as 2.8 μM have been reported (Brandhorst, 1958; Carlucci et al., 1970; Dore and Karl, 1996; GLODAP, V2;
Wada and Hattori, 1972). While in some cases nitrite can be present throughout the entire surface water column
(Lomas and Lipschultz, 2006; Zakem et al., 2018), nitrite concentrations are below detection in most of the ocean
because nitrite is an intermediate formed during oxidation of reduced N ($NH_4^+$, DON) (Zehr and Kudela, 2011; Zehr
and Ward, 2002). The accumulation of nitrite at the PNM occurs at a depth horizon where dynamic N cycling occurs
and can appear and disappear within the span of only 25 meters. The PNM location generally coincides not only with
the top of the nitracline, but also with the depth of the oxycline, the depth of the chlorophyll maximum, and just below
or coincident with an ammonium maximum near the base of the euphotic zone (Dore and Karl, 1996; Herbland and
Voituriez, 1979; Holligan et al., 1984; Kiefer et al., 1976; Zafiriou et al., 1992; Zakem et al., 2018). The consistent
strong spatial relationships between nitrite and both nitrate and chlorophyll concentrations hint at a relationship
between these environmental parameters and nitrite production, but does not provide a clear mechanism.
Because the PNM sits at a depth where many environmental parameters and microbial N transformations are in
transition, determining the exact controls on nitrite accumulation in the PNM remains difficult (Lomas and Lipschultz,
2006; Wan et al., 2021, 2018; Zakem et al., 2018). Near the PNM, the three main microbial groups involved in nitrite
cycling are bacterial and archaeal nitrifiers and phytoplankton. Nitrification comprises the oxidation of ammonia to
nitrate with nitrite as an intermediate. Archaeal ammonia oxidizers dominate the oxidation of ammonia to nitrite
(Francis et al., 2007, 2005; Mincer et al., 2007; Santoro et al., 2010; Schleper et al., 2005), while bacterial nitrite
oxidizers convert nitrite to nitrate (Lücker et al., 2013, 2010; Ward and Carlucci, 1985; Watson and Waterbury, 1971).
Many phytoplankton can also both produce and consume nitrite. Traditionally, phytoplankton are thought to be
consumers of inorganic N, but it is now well documented that they also release inorganic N, including nitrite (Al-
Qutob et al., 2002; Collos, 1998, 1982; Lomas and Glibert, 2000). Nitrification and photosynthesis co-occur near the
depth of the PNM, making it difficult to determine how microbes interact and transform nitrogen to produce the nitrite
concentrations observed, how microbial physiologies differentially respond to gradients in environmental conditions
and ultimately what factors influence the magnitude and depth of the PNM (Ward et al., 1989).
The combination of each microbial group's physiological responses to environmental parameters controls the vertical
profiles of concentrations of different N species and leads to accumulation of nitrite at the PNM. Imbalance between
the two steps of nitrification has been used to explain nitrite accumulation; variations in light levels may cause
differential photoinhibition or differential recovery from photoinhibition of nitrite oxidizers leading to accumulation
of nitrite (Guerrero and Jones, 1996; Olson, 1981). Ammonia oxidizing bacteria are less sensitive to light, have quicker
recovery times to light stress, and are active at higher rates under light stress compared to nitrite oxidizing bacteria



(Guerrero and Jones, 1996; Olson, 1981). Recent studies focusing specifically on the numerically dominant ammonia
oxidizing archaea (AOA), have shown high variation in light tolerance across AOA phylotypes which may explain
the lack of strong light inhibition of ammonia oxidation in some studies (Horak et al., 2018; Merbt et al., 2012; Smith
et al., 2014). Additionally, nitrification rates are substrate-dependent and constrained to places and times when
ammonia and nitrite are both available (Martens-Habbena et al., 2009). Nitrite is also taken up by phytoplankton but
this process is thought to be light dependent (Lomas and Glibert, 2000; Mulholland and Lomas, 2008). Nitrite release
from phytoplankton is also well documented in culture studies (Al-Qutob et al., 2002; Collos, 1998), but it is still
unclear whether nitrite release occurs during incomplete nitrate reduction under low light conditions when energy for
its complete assimilation is limited, under fluctuating high light conditions as a photoprotective mechanism, or as a
stress response (Collos, 1982; Kiefer et al., 1976; Lomas and Glibert, 1999, 2000; Wada and Hattori, 1971).
Accumulation of nitrite occurs when the rate of its production exceeds that of its consumption. Thus, the presence of
the PNM is an indicator of conditions where net production and consumption of nitrite are, or have recently been,
imbalanced (Wada and Hattori, 1971). The accumulation of nitrite in the PNM may provide valuable insight into the
balance of relative rates of microbial nitrite cycling in the upper ocean, as it indicates a zone where biologically
mediated processes are not in balance and may be experiencing differential inhibition or limitation. Rarely are the four
major microbial processes related to PNM formation (ammonia oxidation, nitrite oxidation, nitrate reduction and
nitrite uptake) measured simultaneously in the field. A few studies have measured both steps of nitrification
concurrently near the ETNP (Beman et al., 2013; Füssel et al., 2012; Peng et al., 2015; Santoro et al., 2013; Ward et
al., 1982). The few paired rate measurements that exist tend to show that ammonia oxidation rates exceed nitrite
oxidation rates in the PNM, suggesting nitrite oxidation is the rate limiting step in the reaction pair and a potential
mechanism for nitrite accumulation (Beman et al., 2013; Schaefer and Hollibaugh, 2017). However, the lack of paired
measurements focused on the sharp PNM boundaries limits our understanding of the coupling/uncoupling between
the two steps of nitrification or other processes affecting nitrite accumulation across these depths. Higher resolution
paired measurements will allow us to investigate how environmental gradients create vertical zonation in the relative
rates of nitrite-cycling processes that lead to nitrite accumulation within narrow depth horizons. Previous
investigations of the PNM have typically focused on nitrifier communities or phytoplankton communities separately,
although it is understood that the niches of these communities overlap, and that both may contribute to nitrite
accumulation. The studies that have measured both phytoplankton and nitrifier processes (Mackey et al., 2011;
Santoro et al., 2013; Wan et al., 2018; Ward, 2005) support the idea that physiological constraints and competitive
interactions between these groups drive resource use and nitrite accumulation (Smith et al., 2014; Wan et al., 2021;
Zakem et al., 2018).
Understanding the controls on rates of co-occurring nitrite cycling processes will help clarify the distributions of
microbial activity and how relative rates of these processes may change due to future environmental perturbations.
For example, understanding the controls on and patterns of nitrification in the surface ocean is critical for
understanding new production, as estimates suggest more than 30% of oceanic primary production is supported by
nitrate supplied by nitrification in the euphotic zone (Santoro et al., 2010; Ward et al., 1989; Yool et al., 2007). In





addition, the relative contributions of nitrification and phytoplankton activity to the formation of the PNM may also
be important for understanding the potential for nitrous oxide formation in the surface ocean (Burlacot et al., 2020;
Kelly et al., 2021; Plouviez et al., 2019; Santoro et al., 2011).
In the oligotrophic Eastern Tropical North Pacific (ETNP), the PNM occurs as a sharp peak located at the top of the
nitracline and within the waning slope of the primary chlorophyll maximum. The PNM also sits above a secondary
nitrite maximum (SNM) that occurs within the oxygen deficient waters below. To investigate the relative contributions
of nitrification and phytoplankton processes to net accumulation of nitrite at the PNM feature, we measured rates of
four microbially-mediated nitrite cycling processes (ammonia oxidation, nitrite oxidation, nitrate reduction and nitrite
uptake) in vertical profiles through the PNM. We analyzed spatial and regional variations in environmental conditions
and water column features associated with the PNM, as well as the rates of nitrite production and consumption.
**2 Methods**
**2.1 Hydrography and nutrient analyses**
This study is based on data collected from four cruises to the Eastern Tropical North Pacific between April 2016 and
June 2018 (RB1603 – *R/V Ronald Brown*, April 2016; SKQ201617s – *R/V Sikuliaq,* December 2016; SR1805 – *R/V*
*Sally Ride*, April 2018; and FK180624 – *R/V Falkor*, June 2018; Figure 1). The ETNP hosts one of the largest oceanic
oxygen deficient zones (ODZs) and is a region of active nitrogen cycling. Oxygen concentrations decline precipitously
from saturated surface water concentrations to nanomolar levels across the oxycline in much of the study area (Cline
and Richards, 1972), with oxygen deficient waters beginning as shallow as 15 m at some coastal stations. This study
focused on nitrite cycling in the upper water column near the PNM, and all rate data were collected in oxygenated
waters in or above the oxycline.
Fifty-three stations were occupied during these cruises, and hydrographic observations of temperature, salinity, and
oxygen were made using a CTD package (RB1603 – Sea-Bird SBE 11+ CTD, SKQ201617s/SR1805/FK180624 –
Sea-Bird SBE 911+ CTD). Fluorescence and photosynthetically active radiation (PAR) measurements were measured
at a subset of stations (RB1603 – LiCor Biospherical Photosynthetically Active Radiation Sensor/SeaPoint
Chlorophyll Fluorometer). Discrete water samples were collected from Niskin bottles mounted to the CTD rosette to
measure dissolved inorganic N concentrations. Nitrite and ammonium concentration measurements were typically
made immediately onboard the ship, while samples for nitrate concentration measurements were 0.22 μm filtered and
frozen in 60-ml HDPE bottles for analysis at a shore-based laboratory. During the 2016 cruise, a pump profiling
system (PPS; as described in Codispoti et al., 1991) was also deployed with a separate CTD package (Seabird SBE-
19+, WetStar Fluorometer) at all 16 stations.
For all cruises, nitrite concentrations were measured colorimetrically (Strickland and Parsons, 1972). Briefly, five ml
of sample water from each Niskin bottle was reacted with 200 μl each of sulfanilamide and N-(1-NAPHTHYL)-
ethylenediamine reagents, and absorbance at 543 nm was measured after a 10 min reaction time and converted to
concentration using a standard curve, with an overall precision of ±0.006 μM. Ammonium concentrations were



measured shipboard using a fluorometric method after derivatization with ortho-phthaldialdehyde (OPA) reagent
(Holmes et al., 1999). Samples and standards were reacted with OPA for ~8 hours at 4ºC in the dark before
measurement. In 2016, samples for nitrate plus nitrite were collected from discrete depths using Niskin bottles
mounted to a CTD rosette and analyzed shipboard using an Astoria Pacific autoanalyzer according to the
manufacturer's specifications using standard colorimetric methods (Strickland and Parsons, 1972). In 2017, nitrate
plus nitrite samples were analyzed using standard colorimetric methods on a Technicon Autoanalyzer at the University
of Washington. In 2018, nitrate plus nitrite was measured after Cd reduction using a WestCo SmartChem 200 Discrete
Analyzer at Stanford University, with an overall precision of ±0.6 μM. Nitrate concentrations were calculated by
subtracting nitrite from the concentration of nitrate plus nitrite for all cruises. During the 2016 cruise (RB1603), cast
water from the pump profiling system (PPS) was pumped directly through a Fast Repetition Rate Fluorometer (FRRF)
for chlorophyll *a* fluorescence measurements and then to an Alpkem Astoria-Pacific rapid-flow analysis system for
near-continuous profiles of nitrate, nitrite, and ammonium concentrations at one measurement per second and binned
to every meter (Holmes et al., 1999; Sakamoto et al., 1990; Strickland and Parsons, 1972).
Water column profiles were analyzed to determine station-specific water column features (Table. 1). The depth of the
top of the nitracline ($Z_{nit}$) was identified as the inflection point between the nitrate-depleted surface waters and an
increase in nitrate concentration of 1 μM compared to a reference surface depth of 20 m (Cornec et al., 2021). In
addition, the standard nitracline depth ($Z_{mnit}$) was identified as where the nitrate gradient was steepest. Similarly, the
top of the oxycline ($Z_{oxy}$) was identified as the inflection point between oxygen-replete surface waters, using a
reference depth of 20 m, and a decrease in oxygen concentration of 5 μM. The standard oxycline depth ($Z_{moxy}$) was
where the oxygen gradient was steepest. Other station-specific water column features identified include depth and
concentration of the nitrite maximum (m and μM, respectively), depth and concentration of the chlorophyll maximum
(m and mg m$^{-3}$, respectively), depth and concentration of the ammonium maximum (m and nM, respectively) and the
depth of 1% surface photosynthetically active radiance (PAR) level (m). Concentrations/characteristics of these
variables specifically at the depth of the nitrite maximum were also calculated (e.g., nitrate concentration ($NO3_{pnm}$),
chlorophyll concentration ($Chl_{pnm}$), ammonium concentration ($NH4_{pnm}$), temperature ($T_{pnm}$), density ($D_{pnm}$), percent
of surface PAR ($PAR_{pnm}$), oxygen concentration ($O_{pnm}$). The Brunt-Väisälä frequency ($BV_{pnm}$) was calculated at the
PNM nitrite maximum (±8 m) using the equation $N = \sqrt{\frac{-g}{\rho} * \frac{d\rho}{dz}}$, where $g$ is the acceleration due to gravity (m s$^{-2}$), $z$
is depth (m) and ρ is density (kg m$^{-3}$). Depth-integrated concentrations of nitrate, nitrite and ammonium (μmol N m$^{-2}$)
were calculated for the euphotic zone (upper 120 m), capturing the entirety of the PNM feature.

**Table 1. Water Column Feature Acronyms, Definitions and Units**

| Symbol | Definition | Unit |
|---|---|---|
| PNM | Primary nitrite maximum, whole feature | — |
| $Chl_{max}$ | Concentration of the deep chlorophyll maximum | mg m$^{-3}$ |
| $NH4_{max}$ | Concentration of the ammonium maximum | nM |



| NO2$_{max}$ | Concentration of the PNM nitrite maximum | $\mu$M |
|---|---|---|
| Z$_{chl}$ | Depth of the deep chlorophyll maximum | m |
| Z$_{nh4}$ | Depth of the ammonium maximum | m |
| Z$_{no2}$ | Depth of the PNM nitrite maximum | m |
| Z$_{nit}$ | Depth of top of the nitracline | m |
| Z$_{mnit}$ | Depth of steepest gradient in nitracline | m |
| Z$_{oxy}$ | Depth of the top of the oxycline | m |
| Z$_{moxy}$ | Depth of steepest gradient in oxycline | m |
| Z$_{pPAR}$ | Depth of 1% surface PAR | m |
| Chl$_{pnm}$ | Chlorophyll concentration at the PNM peak | mg m$^{-3}$ |
| NH4$_{pnm}$ | Ammonium concentration at the PNM peak | nM |
| NO3$_{pnm}$ | Nitrate concentration at the PNM peak | $\mu$M |
| T$_{pnm}$ | Temperature at the PNM peak | C |
| D$_{pnm}$ | Density at the PNM peak | kg m$^{-3}$ |
| PAR$_{pnm}$ | Percent of surface PAR at the PNM peak | % |
| O$_{pnm}$ | Oxygen concentration at the PNM peak | $\mu$M |
| BV$_{pnm}$ | Brunt Väisälä Frequency at the PNM peak | s$^{-1}$ |
| NH4_Int | Depth integrated ammonium over upper 120m | nmol N m$^{-2}$ |
| NO2_Int | Depth integrated nitrite over upper 120m | $\mu$mol N m$^{-2}$ |
| NO3_Int | Depth integrated nitrate over upper 120m | $\mu$mol N m$^{-2}$ |
| Chl_Int | Depth integrated chlorophyll over upper 120m | mg m$^{-2}$ |


## 2.2 Nitrite cycling rates

Rates of ammonia oxidation, nitrite oxidation, nitrate reduction and nitrite uptake were measured at 12 of the 53
stations occupied over 4 cruises from 2016-2018 (Fig. 1a); five stations from 2016, two stations in 2017, and five
stations in 2018. At each of these stations during a pre-dawn cast, 3-4 depths near the PNM were sampled based on
real-time CTD fluorescence data during the downcast, targeting depths both within the chlorophyll maximum and on
the upslope and downslope of its peak. When available, nitrite profiles from previous casts were consulted to guide
sampling based on the location of the PNM within the chlorophyll maximum.



Rate measurements for microbial nitrite cycling processes were made using [15]N-tracer incubation experiments. From
each depth, six clear 500-ml HDPE Nalgene bottles were triple-rinsed and filled directly from the Niskin bottle for
light incubations. Additionally, six 500-ml or 1-L amber HDPE Nalgene bottles were triple-rinsed and filled for paired
dark incubations. One of three [15]N-labeled nitrogen substrates (K[15]NO$_3^-$ enriched at 99.5atm%, Na[15]NO$_2^-$ enriched at
98.8atm% or [15]NH$_4$Cl enriched at 99.5atm%) was added to duplicate bottles to achieve enrichments of 200 nM [15]N.
After gentle mixing, a 60 ml subsample was Sterivex-filtered (0.22 μm pore size) using a syringe to determine initial
concentration and [15]N enrichment of the substrate pool. Approximately 10 ml was used for shipboard measurement of
the initial concentrations of total nitrite or ammonium (ambient concentration plus [15]N-labeled DIN addition). The
other 50 ml was frozen in a 60-ml HDPE bottle for measurement of total nitrate concentration and isotopic enrichment.
Each incubation bottle was placed in a deck-board incubator that approximated the ambient light level from the sample
collection depth using neutral density screening. The percent PAR in the incubators was recorded using a submersible
Licor PAR meter or an *in situ* HOBO light and temperature logger (~1%,~4%,~20% surface PAR). Incubators were
plumbed with flow-through surface seawater to maintain in situ water temperatures. However surface water
temperatures were often significantly warmer than those at collection depth and could have biased some of the
incubation results. Subsamples were collected from each incubation bottle after approximately 8, 16 and 24 hrs.
Samples were Sterivex-filtered (0.22 μm pore size) and filtrate frozen in 60-ml HDPE bottles for nutrient and isotope
analysis. At the end of the incubation (24 hr), the remaining ~300 ml of water was filtered onto a pre-combusted
(450°C for > 4 h) GF/F (0.7 μm) filter; the filter was folded and placed into a cryovial and stored at -80°C for later
analysis of particulate [15]N analysis at the University of Hawaii Isotope Lab. All seawater samples were stored frozen
until the time of isotopic analysis. Incubation bottles were acid washed and re-used for experiments using the same
[15]N substrate.

**2.3 Isotope analysis and rate calculations**

For estimates of ammonia oxidation, nitrite oxidation and nitrate reduction rates, samples collected from each
timepoint were analyzed for [15]N enrichment in the product pool for each process (Table 2). For each sample, the
product was converted to nitrous oxide either by bacterial (*P. aureofaciens*) conversion using the denitrifier method
(McIlvin and Casciotti, 2011; Sigman et al., 2001) or chemical conversion using the azide method (McIlvin and
Altabet, 2005). Isotopic analysis via the denitrifier method was used for measurement of [15]NO$_X$ (ie. [15]NO$_3^-$ + [15]NO$_2^-$)
in ammonia oxidation and nitrite oxidation experiments. Measurements of nitrite oxidation required pre-treatment of
samples to remove any remaining [15]N-NO$_2$ prior to analysis of [15]N-NO$_3$ (Granger and Sigman, 2009). Briefly, 10 ml
of each sample was treated with 100 μl of 4% sulfamic acid in 10% hydrochloric acid for 15 min, after which the pH
was neutralized using 85 μl of 2M sodium hydroxide before proceeding with denitrifier method. Samples were
prepared in volumes targeting 20 nmoles nitrate plus nitrite. The azide method was used to prepare nitrite produced
from nitrate reduction experiments for isotopic analysis (McIlvin and Altabet, 2005). Nitrite was converted to nitrous
oxide by incubating for ~30 min with a 2M sodium azide solution in 20% acetic acid. The reaction was neutralized
with 6M sodium hydroxide prior to isotope analysis. Since nitrite product concentrations were low (<2 μM), a
significant portion of the nitrite in the samples was newly created from [15]N-labeled nitrate, thus carrier nitrite (5-10





nmoles) of known isotope value was added to dilute the [15]N enrichment and increase overall concentration of nitrite
in the samples before analysis. Samples were analyzed in volumes targeting 10 nmoles of nitrite.
The isotopic composition of the nitrous oxide produced from each sample was measured in the Casciotti Laboratory
at Stanford University using an isotope ratio mass spectrometer (Thermo-Finnigan Delta[PLUS] XP) fitted with a front-
end custom purge-and-trap gas purification and concentration system (McIlvin and Casciotti, 2011). Each set of 9
samples was bracketed with international reference materials to correct for instrument drift and sample size, and to
calibrate isotope values. USGS32, USGS34, and USGS35 (Böhlke et al., 2003) were used to calibrate nitrate isotope
analyses, and RSIL-N23, N7373 and N10219 (Casciotti et al., 2007) were used to calibrate nitrite isotope analyses.
For nitrate reduction samples, additional mass balance corrections were made to correct for the addition of nitrite
carrier to the product pool before calculation of rates. The denitrifier method for natural abundance nitrate isotope
analyses typically has a precision of better than 0.5‰ for $\delta^{15}N$ (McIlvin and Casciotti, 2011; Sigman et al., 2001),
although standard deviations are higher for isotopically enriched samples. Here, the mean analytical precision of $\delta^{15}N$-
$NO_X$, $\delta^{15}N$-$NO_3$, $\delta^{15}N$-$NO_2$ measurements of [15]N-labeled samples were ±4.2‰, ±4.6‰ and ±0.7‰, respectively,
corresponding to mean coefficient of variance (CV%) of 5.3%, 0.56% and 9.7%, respectively (Table 2).
**Table 2. Nitrite cycling reactant and product pools as analyzed by isotope rate mass spectrometry after conversion to**
**nitrous oxide via denitrifier or azide methods.**

| Microbial Process | [15]N-labeled Reactant | Prep Method | [15]N-labeled Product | Mean Precision (‰) | Mean CV % |
|---|---|---|---|---|---|
| *Ammonia Oxidation* | $NH_4Cl$ | Denitrifier | $NO_3^-/NO_2^-$ | 4.2 | 5.3 |
| *Nitrite Oxidation* | $NaNO_2$ | Sulfamic-treated + Denitrifier | $NO_3^-$ | 4.6 | 0.56 |
| *Nitrate Reduction* | $KNO_3$ | Azide w/carrier | $NO_2^-$ | 0.7 | 9.7 |

Rate calculations were made by tracking the increase in product [15]N over the incubation period (Ward, 1985). For
ammonia oxidation the equation is as follows:
$$V_{NH3} = \frac{\Delta[^{15}NOX]_{t8-t0}}{af^{15}NH3_{t0} \times \Delta t \times 24} \qquad\qquad (1)$$
where $\Delta[^{15}NOX]_{t8-t0}$ is the change in product [15]$NO_X$ concentration between the start of the incubation and the 8 h
timepoint (nM), $af^{15}NH3_{t0}$ is the atom fraction of [15]$NH_3$ substrate available at the start of the incubation period, and
$\Delta t$ is the change in time (hours). The ammonia oxidation rate, $V_{NH3}$, is reported in units of nM N day[-1]. A similar
equation was used to calculate nitrite oxidation and nitrate reduction rates, substituting the appropriate substrate and
product species for each process (Table 1). The 16 h and 24 h time point samples were analyzed but not used to
calculate rates as experiments showed non-linear trends after 8 hours of incubation due to substrate depletion. Based
on a threshold increase in product $\delta^{15}N$ compared to the initial product, a theoretical detection limit was calculated to
estimate the rate which we can reasonably expect to discern from zero (Santoro et al., 2013). This calculation is
sensitive to both the $\delta^{15}N$ of the substrate pool, the concentration of the product pool, and the CV% for $\delta^{15}N$





measurements. The threshold for detectable change in product $\delta^{15}$N was approximated using the maximum CV% for
each experiment. For example, if the standard deviation in replicates for a sample with a $\delta^{15}$N of 25‰ was ±0.6, a
CV% of 2.4% was used as the theoretical detectable difference between initial and final $^{15}$N enrichment in the product
pool. Where available, the maximum CV% for each experimental unit was used to calculate the theoretical limit of
detection for each depth (Table S1). The mean theoretical detection limits for ammonia oxidation, nitrite oxidation
and nitrate reduction were 0.5, 6.9, and 0.9 nM day$^{-1}$, respectively. Experimental bottle duplicates were conducted for
most rate measurements and those standard deviations are reported with the final rate data (Table S1).
Filters from nitrite uptake rate experiments were dried overnight and packed in tin capsules before shipment to the
Biogeochemical Stable Isotope Facility at the University of Hawaii, where samples were analyzed on a Thermo
Scientific Delta V Advantage isotope ratio mass spectrometer coupled to a Costech Instruments elemental analyzer.
Rate calculations relied on $^{15}$N enrichment of the particulate organic nitrogen over the 24 h incubation period as in
Dugdale and Goering (1967). Uptake rates were calculated according to Dugdale and Wilkerson (1986) where the
initial $^{15}$N atom percent fraction of the reactant pool was calculated assuming 0.3663 for the $^{15}$N atom percent of the
ambient substrate pool and 98.8atm% $^{15}$N-NO$_2^-$ of the isotope tracer addition. Nitrite uptake rates may be
underestimated due to dilution of the substrate pool via regeneration over the 24 incubation period, and loss of tracer
to unmeasured DON pools (Bronk et al., 1994; Glibert et al., 2019). No correction was made for possible rate
enhancement due to tracer addition (Dugdale and Wilkerson, 1986).
**2.4 Multiple Linear Regression Analysis**
Multiple linear regression (MLR) models were built to assess the environmental variables that influence the depth and
magnitude of the PNM feature in the ETNP. The first set of MLR models ('full' models) used semi-continuous
measurements (temperature, density, oxygen, chlorophyll fluorescence, PAR, nitrate, nitrite and ammonium) from
CTD/PPS casts collected at 16 stations on the 2016 cruise to predict nitrite concentration. Nitrate, nitrite and
ammonium data were natural-log transformed to satisfy normal distribution assumptions of the multiple linear
regression analyses. Using the R package *leaps*, the model was optimized using a best-subsets selection of the full
variable set to maximize $R^2$ and minimize root mean squared error for each potential model size using 10-fold cross
validation to calculate test error for each sized model (optimization led to selection of 19 variables out of 27 possible
explanatory variables – 7 main and 20 single interactions terms) (Miller, 2020). The model size that minimized test
error was selected, and a best-subsets selection method was used to determine the optimal variable coefficients. MLR
coefficients from the optimized models were then used to predict nitrite concentration for station depth profiles in the
ETNP. Three variations on the 'full' model were made using data from: 1) all stations, 2) a subset of 'coastal' stations
(6, 7, 8, and 9) and 3) a subset of 'offshore' stations (13, 14, 15 and 16). Stations were identified as 'coastal' or
'offshore', based on concentration of the nitrite maxima, concentration of the chlorophyll maxima, and nitracline
depth. Not all stations proximal to the coastline were characterized as 'coastal' (Table S2a, b).
A second set of MLR models ('core' models) was built using a more limited set of core variables from the PPS data
that focused on phytoplankton and nitrifier physiology and metabolism (chlorophyll, nitrate, ammonium, oxygen and





percent PAR). These five environmental variables, their quadratic terms and single interaction terms were included
for 20 parameters in total. This model experiment was constructed to assess the relative importance of these core
variables between 'coastal' and 'offshore' stations; therefore, no model size optimization was used to limit variables.
Instead, optimized coefficients for all variables were determined, and variables that contributed less than 2% of total
$R^2$ in both regional models were discarded. In two cases, a variable that was discarded from one regional model was
added back to keep the variable list identical between both models for ease of comparison. In the coastal 'core' model,
the quadratic term for chlorophyll contributed less than 2% to total $R^2$ but contributed greater than 2% relative
importance within the offshore 'core' model, and was therefore retained in both models. In the offshore 'core' model,
PAR was initially removed during the optimization processes because it contributed less than 2% to model $R^2$, but
was ultimately retained because it contributed greater than 2% relative importance within the coastal 'core' model.
The relative percent importance of each variable was calculated by iterative random-ordered removal of each variable
to estimate percent contribution to total model $R^2$ using the *relaimpo* package in R (Grömping, 2006).

## 3 Results

### 3.1 PNM structure and environmental conditions

The typical PNM feature in the ETNP was a unimodal nitrite accumulation situated just below the chlorophyll
maximum and at the top of the nitracline (e.g., Fig. 1b, c). The PNM feature can be described using characteristics of
the PNM peak (i.e. nitrite maximum (μM) and depth of the nitrite maximum (m)) and an integrated nitrite quantity for
the whole PNM feature. Although nitrite can seasonally accumulate all the way to the surface in some regions (Zakem
et al., 2018), homogenous surface nitrite concentrations were not observed in this dataset. Across the ETNP study
region, stations showed similar relative water column structure in the upper 200 m, although the exact depth and
magnitude of features varied. Generally, the depth distribution of features from shallowest to deepest was the top of
nitracline, the chlorophyll maximum, the ammonium maximum then the Nitrite maximum (Fig. 1b, c). This set of
sequential features occurred near the base of the euphotic zone at most stations. Surface irradiance attenuated through
the water column and the depth of 0.1-1 % surface PAR ranged between 25 m and 150 m depth, with the deepest light
penetration at offshore stations. The chlorophyll maximum was usually found around the 1% surface PAR depth and
within the nitracline. However, there was variation in how deep the chlorophyll maximum sat within the nitracline, as
reflected in the amount of nitrate measured at the depth of the chlorophyll maximum (Table S2a). The depth of the
nitrite maxima tended to occur within the downslope of the chlorophyll maxima, while the rest of the water column
had very low concentrations of both N species. The depth horizon of the PNM was often narrow, with detectable
nitrite concentrations spanning only 30 m in some cases.
The depth of maximum nitrite in the PNM shoaled from an average depth of 103 m at offshore stations to 21 m near
the coast, closely following the shoaling nitracline. In density space, the depth of the nitrite maxima fell within a
narrower range, from 22.1 to 26.3 kg m⁻³, with a mean density across the study region of 24.1 kg m⁻³. The nitrite
maxima had an average concentration of ~600 nM, but a range spanning 60-1520 nM. Two types of stations ('coastal'
and 'offshore') were identified based on water column features. Coastal stations (e.g., 2016 PPS 6, 7, 8, 9) had higher




concentrations of nitrite at the nitrite maxima, shallower depths of the nitrite maxima, more nitrate and slightly more
chlorophyll and light at the depth of the nitrite maxima (Table. S2a). Coastal stations also had shallower oxyclines,
1% PAR depths, ammonium maxima and chlorophyll maxima compared to offshore stations. Depth-integrated
chlorophyll, nitrate and ammonium in the upper 120 m were higher at coastal stations. Offshore stations (e.g., 2016
PPS 13,14, 15,16) had deeper nitraclines, smaller chlorophyll maxima and less light at the depth of the nitrite maxima.
**Figure 1. Map of the ETNP region showing cruise tracks included in this study from four cruises from 2016-2018 (a).**
**Stations where rate measurements were made are marked with white stars. Pump profile data was collected at each station**
**occupied during the 2016 cruise. Mean water column profiles from example 'coastal' stations (8 and 9) and example**
**'offshore' stations (14 and 16) during the 2016 cruise (b, c). Dashed grey line depicts the depth at which dissolved oxygen**
**concentrations declined below 3 µM.**

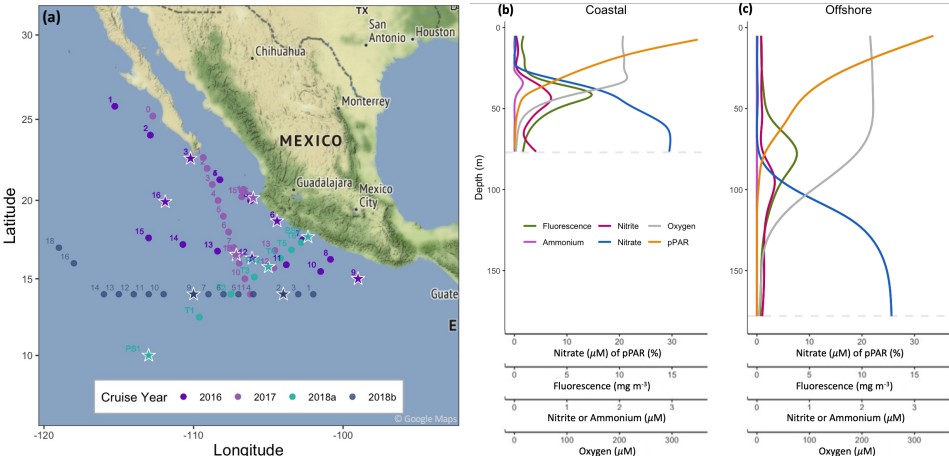

**3.2 Regressions with the nitrite maxima**
There were no strong linear correlations between the concentrations of nitrite and other observed environmental
variables in vertical profiles (chlorophyll, depth, density, oxygen, temperature, nitrate, ammonium) (Fig. S1). This is
unsurprising since the features with unimodal profiles (chlorophyll, ammonium) have concentration maxima that are
offset vertically from the nitrite maxima, and features with other distributions (eg. exponential) are not expected to
have linear relationships with a unimodal nitrite profile. However, spatial relationships between environmental
gradients are still observed in the quantity regressions; for example, the density regression clearly shows that the peak
of the PNM feature consistently fell near 24 kg m$^{-3}$ isopycnal across the region in 2016 (Fig. S1).
To better match unimodal nitrite profiles with spatially offset and vertically non-unimodal environmental gradients,
station-specific features were identified in the high-resolution 2016 PPS profiles and, where possible, in the CTD
datasets (Table 1, e.g., concentration of the nitrite maxima and depth of nitrite maxima). The concentrations of the
nitrite maxima (µM) were regressed against the magnitude of other station features (Fig. 2). The strongest correlation
($R^2 = 0.50$, $p<0.01$) appeared between the concentration of the nitrite maxima and the nitrate concentrations at the
nitrite maxima (Fig. 2c). The Brunt-Väisälä frequency (BV), related to water column stability, also had a strong
positive correlation ($R^2 = 0.40$, $p<0.01$) with the concentration of the nitrite maxima (Fig. 2l). There were weaker



correlations with other parameters such as the concentration of the chlorophyll maxima (mg m$^{-3}$), temperature (°C) at
the depth of the nitrite maxima and oxygen concentration (µM) at the depth of the nitrite maxima ($R^2 = 0.23$, 0.25,
0.29, respectively, all p< 0.04) (Fig. 2a, h, f). The nitrite maxima were not linearly correlated with percent surface
PAR (%) at the depth of the nitrite maxima or the concentration of ammonium (nM) at the depth of the nitrite maxima
(Fig. 2g, d). Depth-integrated chlorophyll, nitrate, and nitrite concentrations in the upper 120 m (excluding ODZ
waters with $O_2 < 3$ µM) were higher when the nitrite maximum was larger (Fig. 2i, j, k). The nitrite maxima did not
correlate with depth-integrated ammonium concentrations (not shown). Inclusion of lower resolution CTD casts from
cruises in 2017/2018 decreased the strength of the linear correlations, likely because of larger error in determining
water column features (e.g., depth of the nitrite maxima, top of nitracline) with larger (~10 m) spacing between discrete
measurements (Fig. S2a).
**Figure 2. Linear regression of concentration of the nitrite maxima against concentrations of water column features,**
**integrated amounts of chlorophyll and DIN, and Brunt-Väisälä frequencies using PPS station data from 2016 (n=16).**
**Chlorophyll concentration at the depth of the nitrite maxima (a), concentration of the chlorophyll maxima (b), nitrate**
**concentration at the depth of the nitrite maxima (c), ammonium concentration at the depth of the nitrite maxima (d),**
**concentration of the ammonium maxima (e), oxygen concentration at the depth of the nitrite maxima (f), percent surface**
**irradiance at the depth of the nitrite maxima (g), temperature at the depth of the nitrite maxima (h), integrated chlorophyll**
**through the top 120m (i), integrated nitrate through the top 120m (j), integrated nitrite through the top 120m (k), and the**
**Brunt-Väisälä frequency across the density gradient ±8m around the depth of the nitrite maxima (l).**

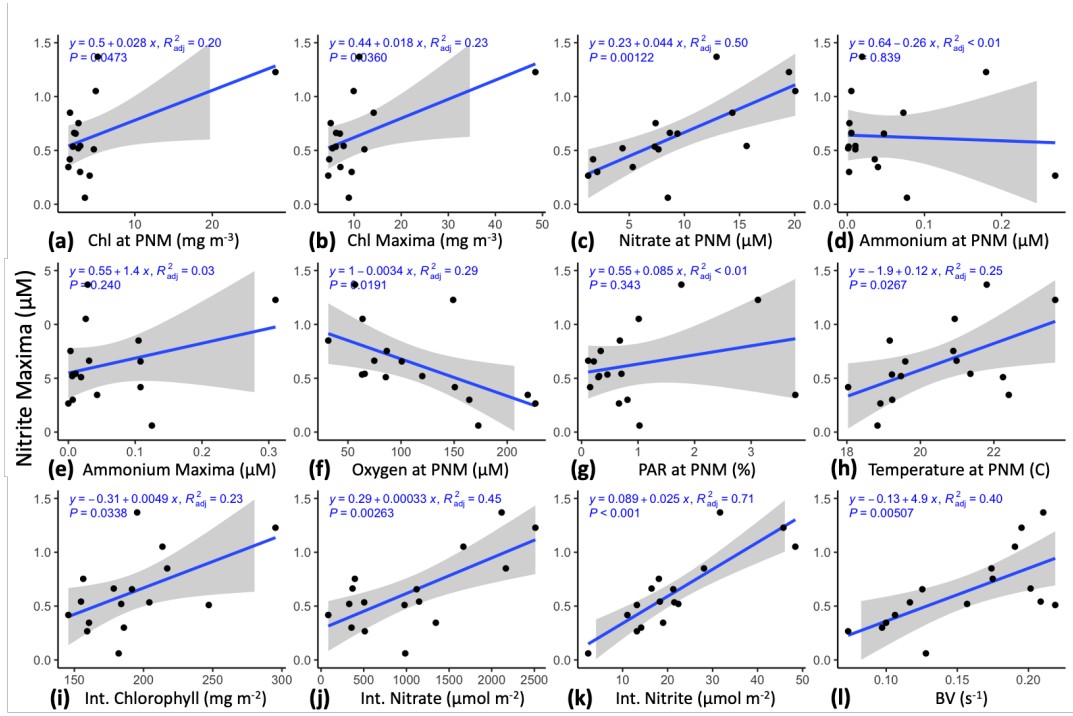


**3.3 Regressions with depth of the nitrite maxima**





The depth of the nitrite maximum at each station was also regressed against the depth of station-specific features (Fig.
3). All water column features showed strong linear correlations with the depth of the nitrite maxima (Fig. 3 a-h). The
top of the nitracline and the depth of 1% PAR had the strongest correlations with the depths of the nitrite maxima ($R^2$
= 0.94, 0.85) (Fig. 3b, g). Correlation of depths of the nitrite maxima with midpoint-calculated oxyclines and
nitraclines was weaker, possibly because those features are less easily defined, or the steepness of these "clines" were
still actively being shaped by the biological responses to changing physical and environmental forcing. The depths of
the nitrite maxima tended to be related to the depth locations of other features and were not as strongly correlated with
the magnitudes of any other feature (Fig. S3). However, the depth of the nitrite maxima and the concentration of the
nitrite maxima were mildly correlated ($R^2 = 0.22$, p = 0.039), with larger nitrite maxima tending to occur at shallower
depths. This correlation became insignificant when the CTD data were included (Fig. S2b). Integrated nitrate had a
strong correlation with the depth of the nitrite maxima ($R^2 = 0.88$, p < 0.01), which is reflective of the depth of the
nitrite maximum tracking with the top of the nitracline. Depth-integrated chlorophyll and nitrite concentrations had
more moderate correlations with the depths of the nitrite maxima ($R^2 = 0.21$, p = 0.041 and $R^2 = 0.32$, p = 0.013,
respectively). Depth-integrated ammonium concentrations did not correlate with the depth of the nitrite maxima.
**Figure 3. Linear regression of depths of the nitrite maxima against water column features from data collected during the**
**2016 cruise using the PPS. Depth of the nitrite maxima was regressed against: a) nitracline depth (m), b) top of the nitracline**
**(m), c) oxycline depth (m), d) top of the oxycline (m), e) depth of the chlorophyll maxima (m), f) depth of the ammonium**
**maxima, g) depth of 1% surface irradiance, h) concentration of the nitrite maxima (μM), i) integrated chlorophyll through**
**the top 120m j), integrated nitrate through the top 120m k), integrated nitrite through the top 120m and l) integrated**
**ammonium through the top 120m. PPS station data from 2016 (n=16).**

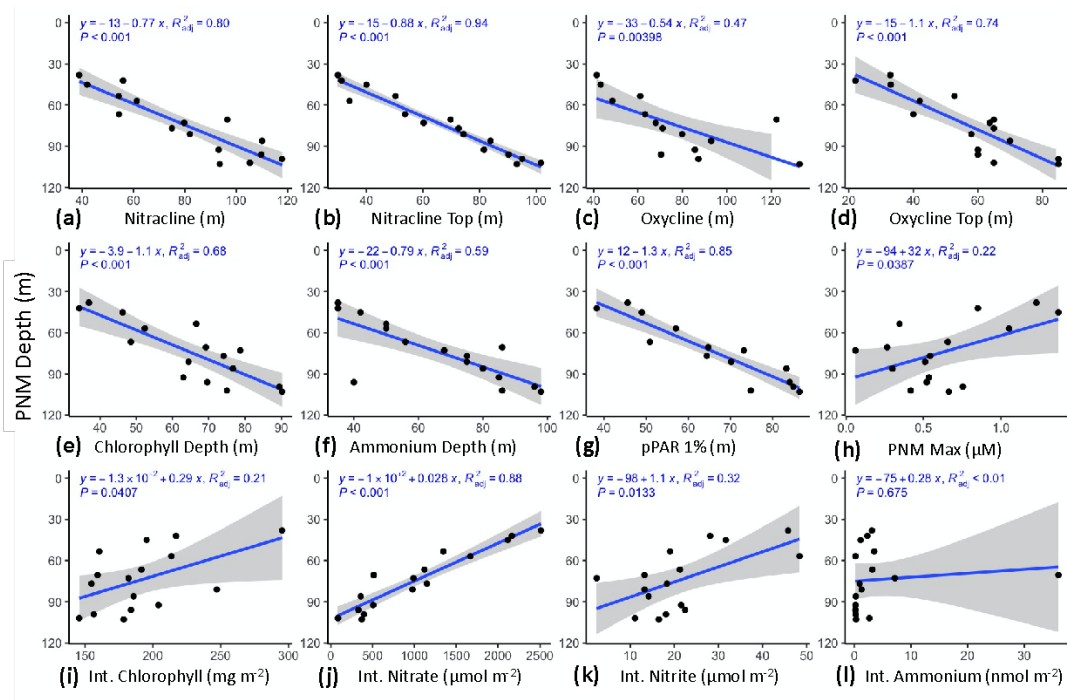






### 3.4 Nitrite Cycling Rates

Rates of nitrite cycling determined for the four major nitrite cycling processes near the PNM were within the same range as previous measurements made in the ETNP region and along the adjacent California coast (Beman et al., 2008; Santoro et al., 2013, 2010; Ward et al., 1982). Within our dataset, the mean rates of ammonia oxidation and nitrite oxidation were similar to each other ($23.7\pm3.5$ and $19.4\pm3.0$ nM day$^{-1}$, respectively), although there was a large range in individual rate measurements across stations and depths, with maximum rates reaching 85.3 and 81.0 nM day$^{-1}$ respectively. Rates of the two phytoplankton-based processes were generally lower and not as similar to each other, with a mean nitrite uptake rate of $19.0\pm5.3$ nM day$^{-1}$ and mean nitrate reduction rate of $6.1\pm1.9$ nM day$^{-1}$. However, nitrite uptake reached one of the highest rates measured at 165 nM day$^{-1}$, and the nitrate reduction rate reached 53.2 nM day$^{-1}$ at a coastal station during the 2017 winter cruise. The pooled mean standard deviation across experimental bottle replicates for ammonia oxidation, nitrite oxidation and nitrate reduction were 3, 4.6 and 1 nM day$^{-1}$, respectively (Table S1).

**Figure 4. Aggregated rate measurements from 2016-2018 with respect to density (sigma T); ammonia oxidation, nitrite oxidation, nitrite uptake and nitrate reduction (panels a-d, respectively) (nM day$^{-1}$), ammonium (nM), nitrite and nitrate (µM), and net nitrite production (nM day$^{-1}$) (panels e-h, respectively), and net consumption, net production, net nitrite production from phytoplankton and net nitrite production from nitrification (nM day$^{-1}$) (panels i-l, respectively). Measurements are colored by relative depth to the station-specific nitrite maximum; above the depth of the nitrite maximum (green crosses), at the nitrite maximum (magenta circles) or below the station-specific depth of the nitrite**



**maximum (blue triangles). The mean ETNP nitrite maxima isopycnal (24.1 kg m$^{-3}$) is marked as a horizontal dashed line**

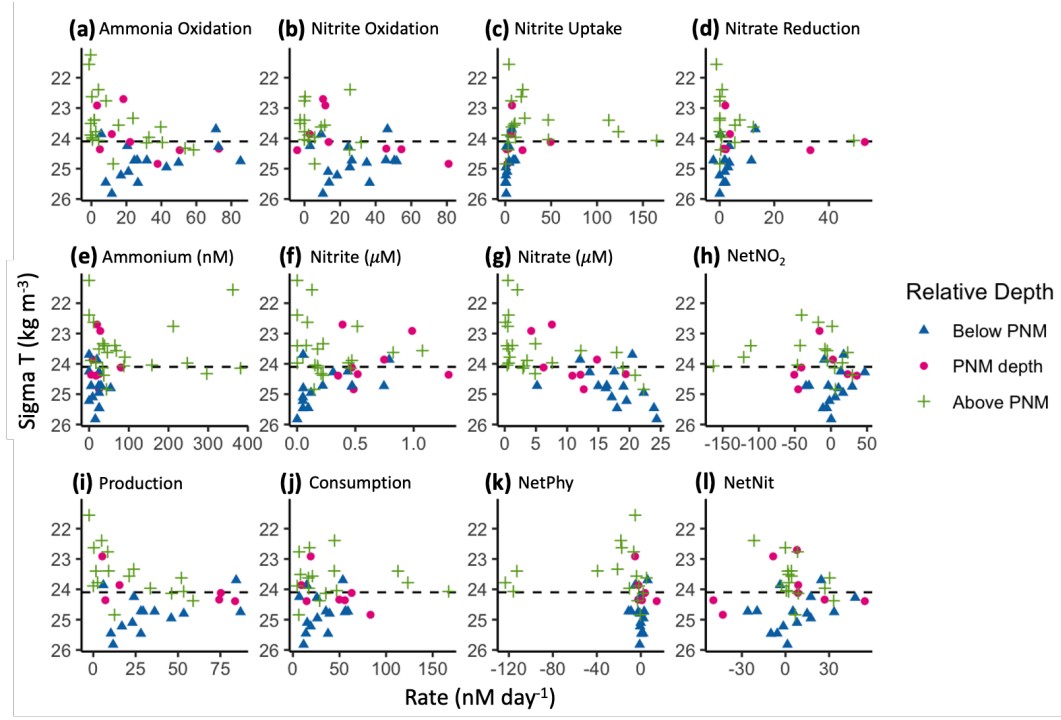


When plotted in density space to aggregate data across years and stations, all processes showed rate maximum at a
subsurface density layer (Fig. 4). Nitrifier processes (ammonia oxidation (Fig. 4a) and nitrite oxidation (Fig. 4b)), had
maximal rates near, or just below, the average density layer for the nitrite maxima across this region (24.1 kg m$^{-3}$).
Nitrite uptake (Fig. 4c) and nitrate reduction (Fig. 4d) rates reached their maxima just above the mean nitrite maxima
isopycnal. Nitrification rates were highest in the lower half of the nitracline, while phytoplankton-mediated processes
(nitrite uptake and nitrate reduction) were highest on the upper slope of the nitracline where nitrite and ammonium
concentrations tended to be higher and light available. While the highest activities of the two microbial groups were
spatially segregated, within-group production and consumption processes had maxima at similar depths. All four rates
formed vertically unimodal distributions, but there was still a large range in measured rates near the peaks with many
rates close to zero.
Net nitrite production from nitrification alone (NetNit = ammonia oxidation – nitrite oxidation) ranged from -49.6 to
54.5 nM day$^{-1}$ with a mean of 5.5±3.1 nM day$^{-1}$ (Fig. 4I). The majority of NetNit values were positive, and maximal
rates were observed just below the mean nitrite maxima isopycnal. Negative NetNit values were driven by high nitrite
oxidation values. Net nitrite production from phytoplankton processes (NetPhy = nitrate reduction - nitrite uptake)
were typically low (mean -13.3±4.9 nM day$^{-1}$), with many negative values resulting from rates of nitrite uptake that
were generally higher than nitrate reduction (Fig. 4k). The largest negative values occurred above the mean nitrite
maxima isopycnal, driven by high nitrite uptake rates where light concentrations were high and nitrate low in the





surface waters. Below the mean nitrite maxima isopycnal, NetPhy remained near zero because both nitrite uptake and
nitrate reduction rates were low. The largest positive NetPhy value was at a coastal station (14 nM day$^{-1}$), where nitrate
reduction reached 33.2 nM day$^{-1}$, but NetPhy was typically an order of magnitude smaller than NetNit.
The vertical distributions of total nitrite production (production = ammonia oxidation + nitrate reduction, Fig. 4j) and
total nitrite consumption (consumption = nitrite oxidation and nitrite uptake, Fig. 4k) showed maximal rates near the
mean nitrite maxima isopycnal (24.1 kg m$^{-3}$). Total nitrite production peaked just below the mean nitrite maxima
isopycnal, with a maximum value of 87 nM day$^{-1}$. Total nitrite consumption peaked just above the mean nitrite maxima
isopycnal, with a maximum value of 167 nM day$^{-1}$. The higher consumption rates just above the mean nitrite maxima
isopycnal were due to higher nitrite uptake rates, especially at coastal stations (Fig. 4c). There was a large range in
rates of nitrite production and consumption processes, but mean values were of similar magnitude (26.4 nM day$^{-1}$ and
39 nM day$^{-1}$, respectively). Total net nitrite production (NetNO$_2$, difference between total production and total
consumption) was highest near the PNM. Negative net nitrite production rates occurred throughout the whole water
column, reflecting high nitrite uptake above the mean nitrite maxima isopycnal and high nitrite oxidation values below
the mean nitrite maxima isopycnal (Fig. 4h). The mean of positive NetNO$_2$ values was 16.7 nM day$^{-1}$ (rates > -2 only,
n=17), although mean NetNO$_2$ was -14.2 nM day$^{-1}$ when all data points were included. The maximum rate of NetNO$_2$
was slightly lower than NetNit alone (46.9 vs 54.5 nM day$^{-1}$, respectively), but the peaks of the vertically unimodal
distributions occurred at the same depths.
While the aggregated rates of NetNO2 peaked near the mean nitrite maxima isopycnal for the region, neither NetNO2
(nor any individual rates) were able to predict the observed nitrite concentrations. Simple linear regressions of each
rate, or calculated net rates, against the quantity of nitrite did not show significance (Fig. S4). Limiting the regression
to a single nitrite maximum and a single highest rate per station also did not show any linear correlation (Fig. S5).
However, some qualitative patterns were noticeable, where the highest rates of phytoplankton-based processes
occurred in samples with lower nitrite concentrations (shallower in the water column). The highest nitrite uptake rates
(>25 nM day$^{-1}$) appeared to restrict the maximum nitrite quantity below 500 nM. Conversely, when high nitrite
concentrations were measured (>600 nM), nitrite uptake rates were low, never higher than 10 nM day$^{-1}$. Nitrate
reduction rates were also higher at lower nitrite concentrations. The highest ammonia oxidation rates (>40 nM day$^{-1}$)
were found where nitrite concentrations were <500 nM (Fig. S4). Nitrite concentrations were highest (>600 nM) where
ammonia oxidation rates were lower (<40 nM day$^{-1}$). The highest nitrite concentrations were associated with waters
having lower nitrite oxidation rates (<20 nM day$^{-1}$), although there were more outliers in this regression. Thus,
although nitrification was an important contributor to total nitrite production, the instantaneous rates could not be used
to predict nitrite concentrations nor could maximum rate at a station predict the nitrite concentration at the nitrite
maxima.
Assuming approximate steady state for PNM nitrite concentrations, rate measurements can be used to calculate a
potential residence time for nitrite across the PNM feature. Using total nitrite production and nitrite concentrations,
the mean residence time was 30.4 days. However, there was a wide range in residence times across all samples,
particularly those from above the average nitrite maxima isopycnal for the region (Fig. S6a). Using total consumption





rates in the calculation gave a slightly lower mean residence time for the region (20.3 days), but again had a large
range in residence times above the mean nitrite maxima isopycnal (0.01-103.2 days) (Fig. S6c). The discrepancy in
residence times calculated using the influx and outflux terms for the nitrite pool suggests that the PNM feature was
most likely not in steady state (as also suggested by the high variation in measured rates at the PNM and inability of
rates to correlate with observed nitrite accumulation), with differences in the dynamics above and below the nitrite
maxima.

### 3.5 Contribution from Nitrification

In considering the metabolisms responsible for accumulation of nitrite at the PNM, it is important to consider the
distribution and magnitude of nitrite production processes vertically through the water column as well as their relative
contributions to total nitrite production. At our sites in the ETNP, ammonia oxidation contributed over 70% of the
total nitrite production through most of the water column (Fig. 5a). The stations where ammonia oxidation contributed
less to total nitrite production were typically coastal stations with low ammonia oxidation rates (e.g., <2 nM day$^{-1}$) or
with high nitrate reduction rates (>20 nM day$^{-1}$). These results support the idea that both ammonia oxidation and
nitrate reduction can contribute to nitrite production, but that the dominant source was from ammonia oxidation at
most stations, particularly at the depth of the nitrite maxima and below. For nitrite consumption, nitrite oxidation
contributed greater than 70% of total nitrite consumption below the mean density layer of the nitrite maxima. Above
the mean density layer of the nitrite maxima, the contribution to total nitrite consumption from nitrite oxidation became
more variable, but with most values below 70% due to more nitrite uptake. Particularly low contributions to total
nitrite consumption from nitrite oxidation were seen above the depth of the nitrite maxima at coastal stations where
nitrite uptake rates were highest. Potential decoupling of ammonia and nitrite oxidation could be seen in the upper
water column, with NetNit peaking at the depth of the nitrite maxima (Fig. 4l), which is more difficult to discern in
the individual ammonia oxidation and nitrite oxidation rates (Fig. 4a,b).

**Figure 5. Percent contribution of nitrification to total nitrite production (a) and total nitrite consumption (b) across density space. Measurements are colored by relative depth to the station-specific nitrite maximum; above the depth of the nitrite**





**maximum (green crosses), at the nitrite maximum (magenta circles) or below the station-specific depth of the nitrite**
**maximum (blue triangles). The mean ETNP nitrite maxima isopycnal (24.1 kg m$^{-3}$) is marked as a horizontal dashed line**

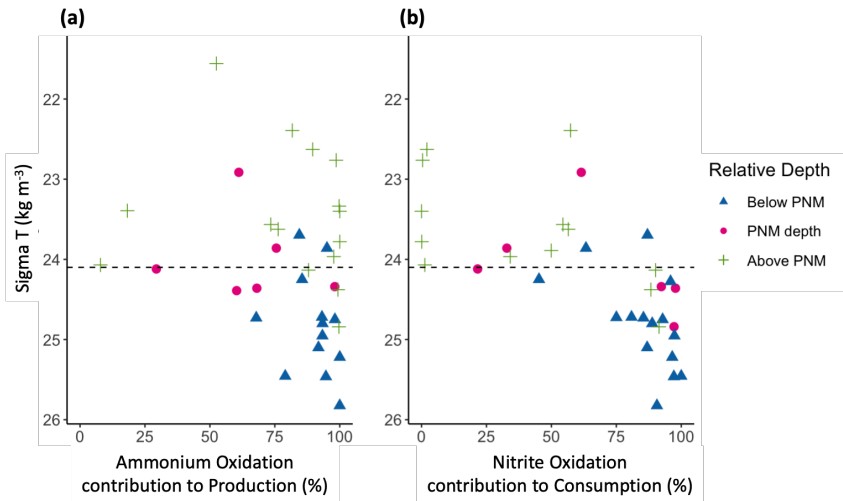


**3.6.1 'Full' model PNM predictions using all stations**


The 'full' model multiple linear regression optimization across all stations resulted in a combination of 10 variables
that were able to predict 66% of the total variance in nitrite concentration (Fig S7a). The final optimized model
included 3 primary variables (chlorophyll, ammonium and oxygen) and 7 interaction terms (Table S3). Based on
relative importance calculations, the temperature-density interaction term contributed the largest amount to the total
variance in nitrite explained by the model (19.8%). The top 3 variables by relative importance all involved
temperature, and in sum contributed 32.3% to the total model $R^2$. Eight out of ten of the variables selected in this
model contributed less than 6% each to total model $R^2$ (Table S3). The predicted vs. observed nitrite slope was less
than 1, meaning small nitrite maxima (<~70 nM) were overpredicted and larger nitrite maxima tended to be
underpredicted (Fig. S7a).
The all-station 'full' model predicted the depth of the nitrite maxima well (mean depth error = 3.7 m) and
underpredicted the nitrite maxima by an average of 230 nM across all stations (when the extreme over-prediction of
15 μM at Station 8 was omitted) (Fig. 6, Table S4). The accuracy of the all-station model varied across station types,
with the most accurate stations including an assortment of coastal, offshore and other (3,4 10,11,13).
**Figure 6. Predicted nitrite profiles from the 'full' MLR models. Observed nitrite (deep pink), all-station model (purple),**
**coastal model (green), offshore model (blue). Coastal stations used for training the coastal model (6, 7, 8, 9) are boxed in**
**green and offshore stations used for training the offshore model (13, 14, 15, 16) are boxed in blue. Note: Scale is adjusted**
**to compare PNM shapes between observations and models, and model maxima may be greater than 1.5 μM. See Table S5**





498        **for model error values and Fig S9 for rescaled profiles.**

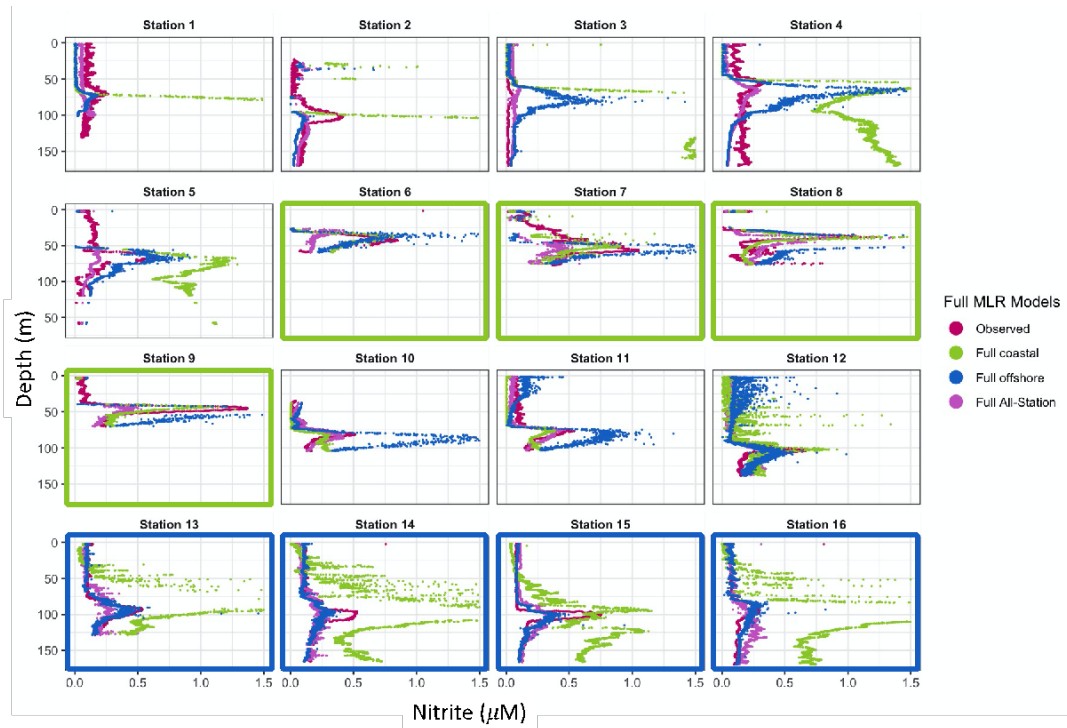


### 3.6.2 'Full' model PNM predictions using coastal stations

For the coastal 'full' model the optimization resulted in 10 variables and was able to predict 77% of the total
variance in nitrite across the coastal stations. The predicted versus observed slope was less than 1, suggesting
slight overprediction of smaller nitrite maxima (<330 nM) and slight underprediction of larger nitrite maxima
(Fig. S7). The most important variable was the nitrate-oxygen interaction term, which explained 17% of the total
model variance. Although nitrate was not included as a primary variable, it was involved in three out of seven
of the interaction terms, and in sum these nitrate interaction terms contributed to nearly half of the total model
$R^2$ (33.8%).
The coastal 'full' model was able to predict the depth of the nitrite maxima well at the coastal stations, with an
average underprediction in depth of only 2.9 m (Fig. 6, Table S4). The concentration at the nitrite maxima at
coastal stations was also accurately predicted by the coastal model, with an average underprediction of only 121
nM. The largest observed nitrite maxima (Station 8) were slightly overpredicted by this model, while the nitrite
maxima at Stations 6 and 7 were well predicted (Fig. 6, green). When applied to non-coastal stations, the coastal
'full' model overpredicted (>2x) the nitrite maxima (except Stations 10,11,12), with an average overprediction
for the whole region of ~1.13 μM (Table S4). The inability of this model to be applied across all stations is



reflected again in the poor correlation between observed and predicted nitrite ($R^2$=0.013, Fig. S7d). The coastal
'full' model predicted a double peak structure at 5 non-coastal stations (4, 5, 10, 13, 15). This PNM shape has
been observed in the field previously (Lomas and Lipschultz, 2006), and was also observed at stations 5 and 10
in our field data. Stations 4, 13 and 15 did not show a double peaked profile, even though it was predicted by the
coastal model.

### 3.6.3 'Full' model PNM predictions using offshore stations

For the offshore 'full' model, 12 variables were included after the optimization process and the final model
explained 79% of the overall variance in nitrite across offshore stations. The predicted vs observed slope was
less than 1, again suggesting slight overprediction of smaller nitrite maxima (<~150 nM) and slight
underprediction of larger sized nitrite maxima (Fig. S7c). The two most important variables in the offshore model
were the oxygen-chlorophyll and density-chlorophyll interaction terms, which each explained 9.4% of the total
nitrite variance (Table S3c). Chlorophyll appeared to be an important parameter in this model, being included as
a primary variable and in 4 interaction terms for a total contribution of 38% to total model $R^2$.
The offshore 'full' model predicted the depth of the nitrite maxima well for offshore stations, with a mean
underprediction in depth of only 0.3 m (Fig. 6, Table S4). The offshore model underpredicted the concentration
of the nitrite maxima by only 53 nM on average. Predicted nitrite profiles accurately captured the concentration
of the nitrite maxima at offshore stations 13 and 16, while offshore stations 14 and 15 were both slightly
underpredicted. The accuracy of the offshore 'full' model applied to all stations was much more variable, with
a mix of fairly accurate (5,12), overprediction (3,4,6,7,8,10,11) and underprediction (1,2) (Fig. 6, Fig. S7). Mean
overprediction of the offshore 'full' model applied across all stations was 855 µM, driven by an extreme
overprediction at Station 8 which when excluded makes the mean size error only 1.23 µM. The offshore model
predicts a slight double peak shape at Stations 5 and 8 only.

### 3.7 Multiple Linear Regression – 'core' model PNM predictions

A subset of 'core' variables was selected and applied in a second MLR analysis in order to directly compare the
influence of each variable between two regions (Coastal vs Offshore stations) (See Methods). There were 7 variables
included in the final 'core' models, with 4 primary variables, 1 quadratic term and 2 interaction terms (Table 2). In
both regional models, nitrate was involved in explaining the most variance (40.8% in the coastal model, 38.8% in the
offshore model).
**Table 2. Coefficients and relative importance from core models; coastal (a) and offshore (b)**





| (a) Coastal 'Core' MLR Coefficients | | | | (b) Offshore 'Core' MLR Coefficients | | |
|---|---|---|---|---|---|---|
| Variable | Coefficient | Percent Importance | | Variable | Coefficient | Percent Importance |
| Oxygen-Nitrate | 0.0028 | 18.9 | | Chl-Nitrate | 0.0752 | 16.7 |
| Nitrate | -0.4137 | 12.2 | | Oxygen-Nitrate | 0.0029 | 11.9 |
| pPAR | -0.0183 | 12.1 | | Chlorophyll | 0.46 | 11.1 |
| Chl-Nitrate | 0.0538 | 9.7 | | Oxygen | -0.0124 | 11 |
| Chlorophyll | -0.0837 | 4.6 | | Nitrate | -0.7093 | 10.2 |
| Oxygen | -0.0047 | 4.6 | | Chlorophyll2 | -0.0994 | 6.8 |
| Chlorophyll2 | -0.0014 | 2.3 | | pPAR | -0.0012 | 4.4 |


### 3.7.1 Coastal 'core' model

The coastal 'core' model was able to explain 83% of the variance in coastal nitrite concentration. The top 3 variables (oxygen-nitrate, nitrate, and pPAR) explained over half of the total model variance (44.1%). Nitrate was the dominant variable, with the combined contribution of all 3 nitrate variables explaining 41.9% of model variance. The total contribution of the chlorophyll related variables was 17.5%. The slope of the predicted vs observed values was 0.71, less than 1, indicating a tendency to overpredict the size of smaller nitrite maxima (< ~350 nM) and underpredict larger sized nitrite maxima (Fig. S8a).

In general, the coastal 'core' model predicted depth well, but was less accurate for nitrite maxima and general shape (Fig. 7). The coastal 'core' model underpredicted the depth of the nitrite maxima at coastal stations (-1.7 m), and underpredicted coastal nitrite maxima by an average of 208 nM, with a large range in error (-830 to +811 nM) (Table S5). Applying the coastal model to the full set of 16 stations showed that the coastal 'core' model overpredicted and underpredicted the nitrite maxima at non-coastal stations, as well as predicting a wide PNM shape that extends deeper in the water column than observed (Fig. 7). The predicted depths of the nitrite maxima from this coastal model fit well with the depth of the observed nitrite maxima, with a mean depth overprediction of only 2.3 m, with a single large outlier at Station 1 where depth was overpredicted by 23.4 m (Fig. 7, Table S5). On average, the coastal model overpredicted the nitrite maxima by just 12 nM across the region. However, there is a wide range in direction of prediction error, with Station 8 being overpredicted by 810 nM and Station 9 underpredicted by 830 nM.

### 3.7.2 Offshore 'core' model

The offshore 'core variable' model was able to explain 98% of the total variance in nitrite (natural-log transformed) at all offshore stations. The relative importance calculation of each variable shows that nitrate, chlorophyll and oxygen together explain more than a third of the total model variance (32.3%). These 3 single variables were also relatively similar in their individual contributions to total model variance (10.2%, 11.1%, 11%) (Table 2). However, nitrate was involved in both interaction terms, which were the top 2 most important variables. The combined contribution of all nitrate effects was 38.8%, almost half of the total model $R^2$ and similar to the coastal 'core' model. The total contribution of chlorophyll related variables is 34.6%, higher than that seen in the coastal 'core' model. The slope of predicted vs observed nitrite was less than 1, indicating a tendency to overpredict the size of smaller nitrite maxima and underpredict larger sized nitrite maxima with a cross-over point between over- and underprediction occurring at ~150 nM nitrite (Fig. S8).





In general, the offshore 'core' model predicted the depth of the nitrite maxima well, but less accurately predicted the
concentration of the nitrite maxima (Fig. 7). The depth of the nitrite maxima at offshore stations was overpredicted by
an average of 2.8 m, and the nitrite maxima was underpredicted by 82 nM at offshore stations. Applying the offshore
model across all 16 stations showed that the offshore core model tended to underpredict the nitrite maxima across the
region. The predicted depth of the nitrite maxima was an average of 5.5 m deeper than the observed depth of the nitrite
maxima, with a range in over and underpredictions from 18.6 m to 5.5 m respectively. The predicted concentration of
the nitrite maxima was lower than observations by an average of 218 nm across the region (Table. S4).
**Figure 7. Predicted nitrite profiles from 'core' coastal MLR (green) and offshore MLR (blue). Observed nitrite profiles**
**from PPS 2016 dataset (deep pink).**

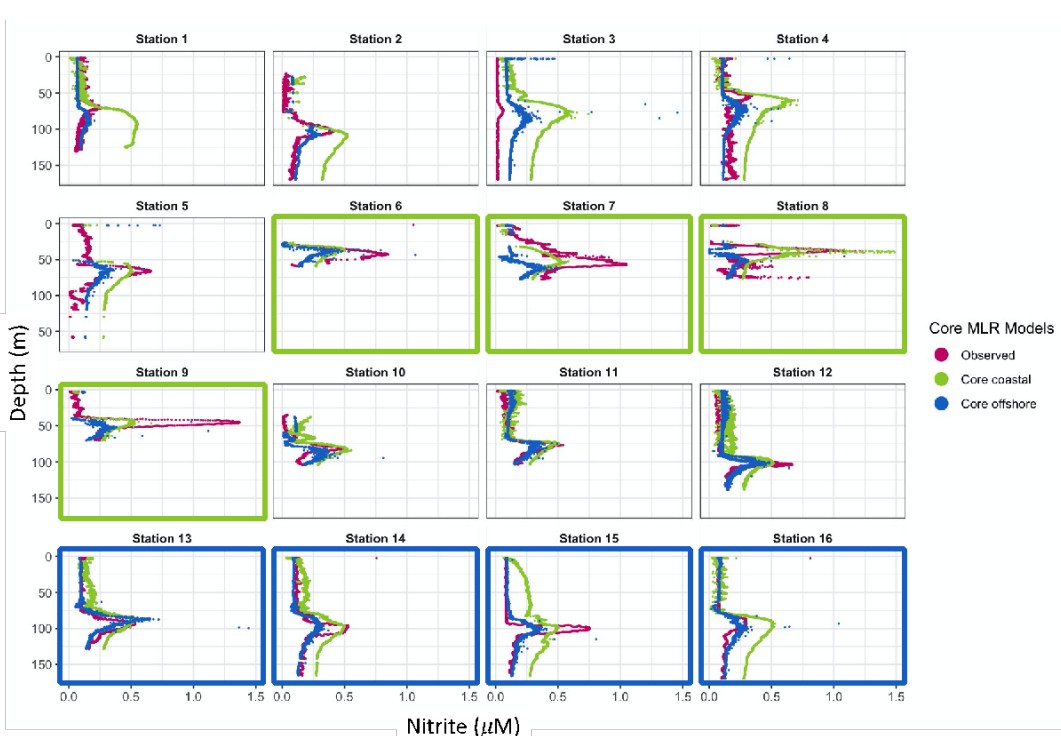





## 4 Discussion

### 4.1 Vertical structure of nitrite accumulation

The same vertical sequence of water column features was seen at all ETNP stations, with the chlorophyll maximum lying above the ammonium maximum lying above the depth of the nitrite maxima at the top of the nitracline. These consistent spatial relationships between water column features suggest that there is a specific set of environmental conditions and biological agents that lead to the accumulation of nitrite. Linear regressions between depth of the nitrite maxima and the depth of other key water column features indeed showed strong correlations. Previous work has noted the connection of the depth of the nitrite maxima with the nitracline (Dore and Karl, 1996; Herbland and Voituriez, 1979; Lomas and Lipschultz, 2006; Meeder et al., 2012; Shiozaki et al., 2016; Vaccaro and Ryther, 1960) and with the chlorophyll maximum (Collos, 1998; French et al., 1983; Kiefer et al., 1976; Meeder et al., 2012), showing that these relationships are shared across multiple oceanic regimes. The environmental feature that correlated most strongly with the depth of the nitrite maximum in our dataset was the top of the nitracline, with the depth of the chlorophyll maximum, the depth of the ammonium maximum, the depth of 1% PAR and the top of the oxycline also having strong correlations, as illustrated by regression analysis (Fig. 3).

The strong covariance between multiple features provides some insight into the mechanisms that link the depth of the nitrite maximum to the environment. Nitrite-cycling microbes respond to the differences in environmental conditions above and below the nitrite maximum. In oligotrophic waters, such as those in the offshore ETNP, uptake of nutrients by phytoplankton maintains low levels of DIN in the upper euphotic zone as physical resupply is low. As light decreases with depth in the water column, active phytoplankton growth is diminished and ammonium and labile dissolved organic nitrogen are released due to grazing and decomposition, providing the low-light, ammonium and reduced organic N conditions suitable for ammonia oxidation. Nitrite oxidizers utilize nitrite produced predominantly from ammonia oxidation to return nitrate to the system. Above the PNM, where light is available, there is enhanced potential for nitrite uptake by phytoplankton and nitrite does not accumulate. Below the PNM, there is diminished supply of ammonium and nitrite oxidizers continue to consume low levels of nitrite produced through ammonia oxidation. At the depth of the nitrite maximum, production terms outweigh both spatially segregated loss terms - nitrite uptake and nitrite oxidation.

The production of nitrite at the PNM is linked to the vertical sequential structuring of the upper water column qualities and is both directly and indirectly dependent on phytoplankton activity. It is directly related via the potential for phytoplankton to release nitrite under varying nitrate supply and light conditions, and indirectly through ammonium supply provided to the ammonia-oxidizing community. Interestingly, the sequence of events that structures the nitracline at the base of the euphotic zone (nitrate and light availability -> uptake of nitrate and phytoplankton growth -> formation of the nitracline and oxycline = release of ammonium (and nitrite) -> oxidation of ammonium by nitrifiers) is ordered similarly to the strength of the linear relationships with depth of the nitrite maximum (top of nitracline > %PAR > chlorophyll/ oxycline > ammonium peak depth). The physical processes that change light and mixing environments initiate the conditions under which phytoplankton and nitrifiers establish their contributions to





the PNM over time. The importance of the time component may help explain why there is variation in the strength of
correlation between instantaneous environmental measurements and a PNM structure that may have been forming
over weeks. Under more dynamic conditions (e.g, coastal upwelling), our observations are more likely to capture a
larger range in scenarios, from initial upwelling to cessation of upwelling, making correlations between depth of the
nitrite maximum and water column features weaker.

### 625 4.2 Concentration of the nitrite maximum

While the depth of the nitrite maximum is predictable based on features of the water column, the concentration of the
nitrite maximum was more challenging to predict. In regressions of water column features against the concentration
of the nitrite maximum, only the amount of nitrate at the nitrite maximum, the Brunt-Väisälä frequency and the amount
of oxygen at the nitrite maximum had moderate linear relationships ($R^2$=0.5, $p < 0.01$, $R^2 = 0.4$, $p = 0.016$, $R^2 = 0.29$,
$p = 0.019$), while the $R^2$ values for the other regressions were smaller ($R^2 < 0.25$) (Fig. 2). The connection between the
nitrite maximum and nitrate concentration may reflect the sequence of events that structures the water column and
forms the nitracline (described above). The presence of increased amounts of nitrate at the depth of larger nitrite
maxima suggests that the phytoplankton have yet to deplete nitrate completely, and a large nitrite maximum is
developing during active phytoplankton nitrate uptake at early bloom formation (Collos, 1982; Meeder et al., 2012).
At a station with a large nitrite maximum, there are also higher concentrations of nitrate at the chlorophyll maximum,
although the chlorophyll maximum may still be small (ie. early bloom).  During this time, ammonium production from
degrading and grazed phytoplankton as well as ammonia oxidation to nitrite may co-occur. Under these early bloom
conditions there is potential to accumulate more nitrite due to increased rates of phytoplankton nitrate reduction, high
rates of ammonia oxidation, and/or decrease in loss terms. Controls on the nitrate reduction rate, and the potential for
ammonium competition interactions between phytoplankton and ammonia oxidizers at nitrate replete depths will be
discussed in relation to nitrite cycling rates.
The linear correlation between the larger nitrite maxima and stronger density gradients (higher Brunt-Väisälä values)
suggests that decreased loss of nitrite via mixing could contribute to larger accumulation of nitrite at the maximum.
However, degradation of the nitrite maximum by mixing would only move existing nitrite away from the peak depth,
not remove it entirely from the water column.
We took two further approaches to understand the correlative disconnect between environmental conditions and nitrite
maxima, 1) polynomial multiple regression analyses which allow multiple variables to co-explain the depth and
concentration of the nitrite maxima, and 2) making direct measurements of the microbial processes that
mechanistically link environmental conditions to the nitrogen transformation rates leading to nitrite accumulation.

### 650 4.3 Predicting nitrite profiles from environmental dataset

The lack of strong linear correlation between nitrite maxima and any single feature may indicate that multiple
conditions need to be met to produce large accumulations of nitrite. For example, other work has shown the largest



seasonal nitrite maxima occur at the onset of the deep chlorophyll maximum, where multiple conditions are met - light
is available and nitrate concentrations are still high (Mackey et al., 2011; Meeder et al., 2012).
Allowing for multiple environmental conditions to contribute, the 'full' multilinear regression models are qualitatively
able to capture the peak shape of the PNM feature using the variables provided, yet are unable to fully explain nitrite
concentration (Fig. 6). For example, the all-station 'full' model explained 66% of the overall variance in nitrite
concentration, and the mean error in nitrite maxima predictions was 740 nM with a large range in errors across stations
(-0.84 to 15.28 μM) (Table S4). This is not surprising, since environmental conditions vary across the ETNP,
especially between coastal and offshore stations. The coastal and offshore nitrite maxima were typically found at
similar densities (~24.1 kg m$^{-3}$ coastal, ~24.3 kg m$^{-3}$ offshore), but at coastal stations the average depth of the nitrite
maxima was 43 m shallower, the average nitrate concentration was 3x higher, the average chlorophyll concentration
was 3x higher, average light was 3x higher, oxygen was 25% higher and ammonium concentrations were also higher
(Table S2). This suggests that the nitrite maxima at coastal and offshore type stations may be innately different, and
possibly controlled by different mechanisms. Taking subsets of station data to make separate coastal and offshore
'full' models allowed for better explanatory power compared to grouping all of the stations together in a single MLR
model (Fig. S7). Model optimization selected different sets of variables to explain the nitrite concentrations in each
model, but nitrate was critical across all three models, aligning with results from the simple linear regression analyses,
where nitrate is important for explaining both depth of the nitrite maximum and the concentration of the nitrite
maximum. While the maximum nitrite concentration was not predicted well by these models, the mean error for the
depth of the Nitrite maximum was less than 4 m for all three 'full' models. However, the predicted depth of the Nitrite
maximum at individual stations could be more significantly erroneous (Table S4).
The 'core' models limited variables to those that had strong single linear regressions with depth and concentration of
the nitrite maxima, and both the coastal and offshore models explained similar amounts of the total variance in nitrite
concentration in their respective regions. Even though both models explained relatively similar amounts of variation
in nitrite concentration and used the same limited suite of variables, different coefficients led to differing predicted
nitrite profiles across stations (Fig. 7, Table 2). In the coastal region, the primary model components included nitrate
and light, two environmental conditions that are related to physical initiation of bloom conditions. The offshore model
shifts importance slightly towards a stronger chlorophyll component and reduces the importance of light.
The nitrate variables were involved in explaining similar amounts of the nitrite variance in both models (coastal and
offshore, 40.8% and 38.8%). Nitrate as a single variable also explained a similar portion of the total model variance
in both the coastal and offshore models (12.2%, 10.2% respectively). The coefficients for nitrate variables have the
same sign in both models, with nitrate having a negative coefficient and the two nitrate interaction terms having
positive coefficients. Overall, the negative nitrate coefficients act to decrease predicted nitrite below the nitrite maxima
where nitrate increases towards ~25 μM. The slightly more negative coefficient in the coastal model is counteracted
by the slightly higher concentrations of nitrate seen at coastal nitrite maxima. The oxygen and pPAR coefficients for
both models are also negative, and act to decrease predicted nitrite at the depths above the nitrite maximum. The
interaction terms containing nitrate in both models have positive coefficient values, adding nitrite to depths near the





PNM where nitrate, oxygen and chlorophyll are all present together. In both 'core' models, the interaction term
between nitrate and chlorophyll is an important variable (>10% $R^2$ in both). This suggests that nitrite accumulation
occurs at depths where chlorophyll, nitrate and oxygen co-exist, corroborating the findings from that linear regression
analyses, that the depth of the chlorophyll maxima, nitracline top and oxycline top are individually important in
determining the depth of the nitrite maximum.
The chlorophyll variables are the only coefficients that differ in sign between the two models, with the coastal
chlorophyll coefficient being negative and the offshore chlorophyll coefficient being positive. The quadratic term for
chlorophyll has a negative sign for both models, meaning the presence of a chlorophyll maximum decreases nitrite
predictions strongly just above the nitrite maximum (perhaps driven by nitrite uptake) and shifts the nitrite peak
towards the downslope of the chlorophyll maximum. The single chlorophyll term in the coastal model is also negative
and reduces nitrite predictions in direct proportion to the size of the chlorophyll peak. In contrast, the positive single
chlorophyll term in the offshore model means that, opposing the quadratic term, this variable adds nitrite at depths
across the chlorophyll maximum. Additionally, the single chlorophyll term in the offshore model is much larger in
absolute magnitude than the coastal term, which likely explains the poor performance of the offshore core model at
coastal stations where chlorophyll concentrations are often larger (Fig. 7). The smaller chlorophyll coefficients used
to model nitrite maxima at coastal stations make the model less sensitive to large changes in chlorophyll, while the
larger offshore coefficient suggests that small changes in chlorophyll offshore have more influence over the resulting
nitrite predictions.
**4.4 Rates of Nitrite Cycling**
Strong single variable correlations with depth of the nitrite maxima and mild correlations with concentration of nitrite
at the nitrite maxima (with supportive findings from the MLR analyses), suggest that while the PNM feature is
consistently limited to specific depths, the concentration of the nitrite maxima may be modulated by more nuanced
environmental timings and microbial physiologies. The two main mechanistic explanations for nitrite production at
the PNM involve the microbial physiology of phytoplankton and nitrifying bacteria/archaea. The overlapping habitats
and competition for DIN resources requires that we consider both microbial groups in our understanding of PNM
formation (Lomas and Lipschultz, 2006; Mackey et al., 2011; Smith et al., 2014; Wan et al., 2021, 2018; Zakem et
al., 2018). This dataset is particularly unique because we have directly measured four major nitrite cycling rates from
the same source water, allowing comparison of relative rates of each process within a community and enabling the
calculation of net rates of nitrite production around the PNM feature. Our expectation at the beginning of this study
was that higher rates of nitrite production, or net nitrite production, would correspond to larger accumulation of nitrite.
Our findings, however, revealed a more complex pattern where the instantaneous rates of gross or net nitrite
production did not reflect the amount of observed accumulated nitrite.
The spatial distribution of measured rates through the water column showed peaks in each process near the PNM,
but with slight variation in where the rate maxima fell relative to the nitrite maxima. The highest phytoplankton activity
was located just above the PNM peak, while nitrification rates were highest near the PNM peak and distributed in a





peak shape, a distribution seen in other nearby systems (Beman et al., 2012; Santoro et al., 2013). Although the
aggregated data from the region showed these spatial segregations by microbial group, this was not always observed
at an individual station. The highest rates of nitrification may be slightly skewed towards the lower slope of the PNM,
but the depth of the nitrite maximum at many stations was determined from discrete measurements taken at ~10 m
resolution, so it is possible that the real maxima occurred between sampled depths. The PPS data allowed much more
precise determination of the depth and peak size, although rate measurements were still limited to lower resolution
sampling.
The vertical distribution of nitrification has been theorized to be controlled by light inhibition, restricting nitrification
to depths at the base of the euphotic zone (Olson, 1981). However, active nitrification has been observed in the sunlit
surface ocean (Shiozaki et al., 2016; Ward, 2005; Ward et al., 1989), leading to new theories suggesting that ammonia
oxidation is restricted to near the PNM not only because of inhibition at high light levels, but because ammonium or
nitrate availability shifts the competitive balance for ammonium acquisition away from phytoplankton and towards
ammonia oxidizers (Smith et al., 2014; Wan et al., 2018; Xu et al., 2019). In this dataset, we did measure nitrification
rates >2 nM day$^{-1}$ at light levels of 25-30% surface PAR at coastal stations, although there was a clear enhancement
of nitrification rates at light levels below 5% surface PAR. Although linear regressions of ammonia oxidation rate
didn't show a strong correlation with the nitrite maxima or depth of the nitrite maxima, there was a relationship
between ammonia oxidation and both nitrate and light (Fig. 8). Similar to the data compiled in Wan et al. (2018), the
highest ammonia oxidation rates were restricted to depths with higher nitrate concentrations and lower light levels.
However, even when constraining the ammonia oxidation rate data to where there is both low light and higher nitrate
concentrations, measurements spanned the entire range of rates from 0-85 nM day$^{-1}$, indicating that the conditions
controlling the depth of the rate maxima do not guarantee high rates, but simply facilitate the possibility of high rates.


**745** **Figure 8. Relationship between nitrite cycling rates and percent surface PAR (a) and nitrate concentration (b).**
**746** **Phytoplankton processes are in green and nitrifier processes are in orange. Nitrite consumption processes are x shapes**
**747** **and production processes are filled circles.**

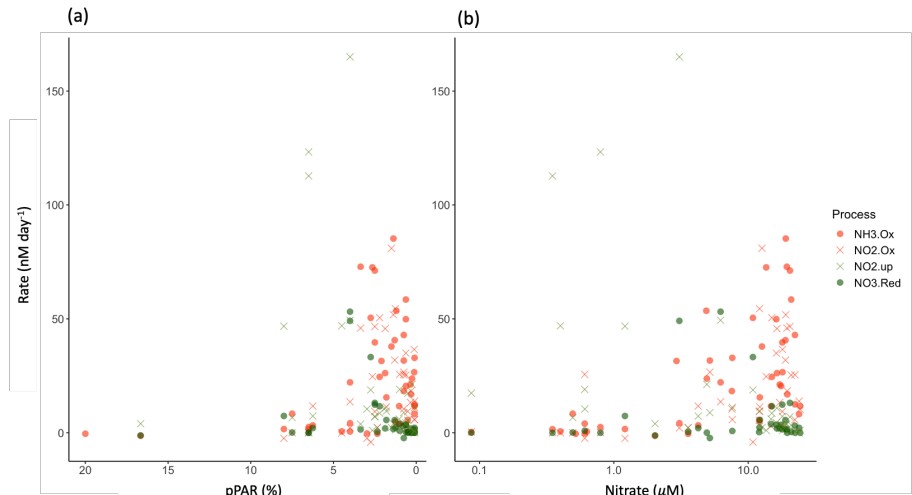

**748**

**749** The individual rate measurements were not correlated with the amount of nitrite accumulated in the water column at

**750** a given depth (Fig. S4). Neither were the net rates (NetNit, NetPhy, NetNO2) able to explain observed nitrite

**751** concentrations. Although the vertical pattern in net nitrite production rates (NetNO2) showed a peak shape that was

**752** qualitatively similar to nitrite concentration, there was no linear relationship between NetNO2 and nitrite

**753** concentration (Fig. 4h, Fig. S4), suggesting that instantaneous in situ rate measurements are not able to capture time-

**754** integrated nitrite accumulation in the PNM. Modeling efforts that are able to integrate fluctuations in environmental

**755** influence on microbial rates over longer time scales may be more able to explain observed nitrite concentrations.

**756** Nitrification rates were similar in magnitude between coastal and offshore stations (Table S2), with the major

**757** differences in rates measurements between coastal and offshore stations found in the phytoplankton processes (nitrate

**758** reduction and nitrite uptake). The highest rates of phytoplankton activity were at coastal stations and occurred

**759** primarily above the depth of the nitrite maximum. The distribution of measured activity lends support to the hypothesis

**760** that phytoplankton may outcompete nitrifiers for DIN sources above the nitrite maxima (Wan et al., 2018; Zakem et

**761** al., 2018). This proposed mechanism accounts for the correlations seen between lower light levels and higher ammonia

**762** oxidation because the top of the nitracline itself is a physical demarcation of the depth where phytoplankton co-

**763** requirements for light and nitrate are met. Previous work has also shown that the presence of nitrate can inhibit nitrite

**764** uptake by phytoplankton through competitive interactions (Eppley and Coatsworth, 1968; Raimbault, 1986) (Fig.8b).

**765** This mechanism may provide a way to connect the presence of nitrate with a larger PNM that relies on prevention of

**766** nitrite loss, rather than an increase in nitrite production.

**767** An additional loss term that could influence the size of the observed nitrite peak may be diffusion, moving nitrite away

**768** (both up and down) from the depth of maximal net nitrite production. The coastal stations were grouped based on the





presence of shallow nitraclines and shallow chlorophyll maxima depths, as well as larger chlorophyll maxima and
nitrite maxima. In addition to these commonalities, the coastal stations also had the steepest density gradients near the
observed larger PNM, making Brunt-Väisälä (BV) frequency correlate with the nitrite maxima in this dataset
(p=0.005) (Fig. 2l). The strong density gradients at the coastal stations (6,7,8,9) would inhibit mixing, potentially
allowing for larger concentrations of nitrite to accumulate where gradients are steepest. This lack of diffusive loss at
coastal stations could partially explain why ammonia oxidation rates can remain similar between coastal and offshore
stations ($25.8\pm3.6$ vs $21.3\pm3.3$ nM day$^{-1}$), yet result in higher accumulated nitrite at an coastal PNM.

**4.5 Different time scales inherent to observational patterns**

Environmental features may not explain the concentration of the nitrite maxima because of a time lag between
environmental conditions measured at a station and the response of the microbial community, and the length of time
needed to produce a PNM. Previous work has shown that a seasonal PNM can develop over 6 days in the Gulf of
Aqaba (Mackey et al., 2011). In our study, a large range was observed in net production rates (~0-86.9 nM day$^{-1}$),
leading to the potential for a PNM to develop in less than a day at some locations, and months at other stations.
However, an imbalance in nitrite production and consumption merely tells you whether nitrite is currently increasing
or decreasing, not whether the instantaneously observed nitrite concentration should be high or low. Calculations from
rate estimates and isotope data suggest 20-50 day residence times for nitrite in the Arabian Sea PNM (Buchwald and
Casciotti, 2013). Ammonia oxidation measurements from the California Current System suggested an 18-470 day
residence time for offshore stations, and 40 day residence time for a coastal station (see full table in Santoro et al.
2013). Our residence time calculations, on the order of days to months, are consistent with these estimates of PNM
residence times (Fig. S6). The high variability in accumulation times across sample locations makes it less likely that
snapshots of rates and environmental conditions would be representative of conditions for a PNM developing over
longer timescales.

**4.6 Spatiotemporal controls on the nitrite maximum**

Previous work investigating the onset of the classic PNM has shown that nitrite concentrations are highest at the
beginning of seasonal stratification when phytoplankton begin to bloom, suggesting that phytoplankton help provide
the necessary conditions for nitrite accumulation (Al-Qutob et al., 2002; Mackey et al., 2011; Meeder et al., 2012;
Vaccaro and Ryther, 1960). In Mackey et al. (2011), the onset of stratification initiates a phytoplankton bloom that
begins to deplete surface nitrate and releases ammonia via phytoplankton degradation and zooplankton grazing. An
accumulation of ammonium forms just below the chlorophyll maximum, which is subsequently followed by an
accumulation of nitrite just below the ammonium peak. This continued stratification pattern supports the persistence
of the emergent PNM feature, though the size of the nitrite maximum declines over the duration of the stratification
period. The correlation between coastal upwelling and higher nitrite accumulation in the ETNP PNM may be
controlled by similar mechanisms as the high nitrite accumulation at the onset of seasonal stratification in other
regions. Instead of a strongly seasonal onset of stratification, the ETNP stratification persists year-round but is
modulated by upwelling along the coast.



At coastal stations in 2016, we saw higher average concentrations of nitrate (16µM) at the depth of the nitrite maxima
due to upwelling conditions, while average nitrate concentrations at offshore nitrite maxima were lower (5.9 µM). The
positive correlation of nitrate concentration at the PNM peak with the concentration of the nitrite maximum ($R^2$=0.5,
p=0.01) suggests that upwelling nitrate is critical for larger nitrite maxima. The correlation found in the MLR analysis
between the chlorophyll-nitrate interaction term and the nitrite maxima supports the idea that higher nitrite
accumulation requires the presence of higher levels of nitrate within the chlorophyll bloom. High variation in the
correlation of nitrite maxima with chlorophyll, ammonium and nitrate may be due to how recently the chlorophyll
bloom was initiated (has the bloom had time to draw down available nitrate?). However, these patterns do not identify
whether the presence of nitrate drives nitrite production from phytoplankton directly, or indirectly by stimulating
ammonia oxidation, or is a proxy for a more stratified water column.
Sequential decomposition of particulate organic nitrogen (PON) produces ammonium, then nitrite, and nitrate over
time, and matches the spatial ordering of these species with depth in the water column (Meeder et al. 2012). In a
stratified water column, the vertical transport of material may be slow enough to allow for a similar temporal
degradation pattern to emerge across the pycnocline. The sequence is initiated by the blooming of phytoplankton,
which is restricted to surface depths with adequate light and nitrate. In a coastal upwelling regime, the stratified water
column is pushed up towards the surface, and this degradation sequence is modified by enhanced source PON from
larger chlorophyll blooms. Larger pools of chlorophyll lead to larger ammonium and nitrite accumulation. Based on
the magnitude of net nitrite production attributed to phytoplankton vs. nitrifiers, nitrifiers appear to have a larger
potential for net nitrite production at ETNP PNMs. The association of nitrification rates with increasing nitrate
concentration, which is not a required substrate for nitrification, indicates an indirect connection with phytoplankton
activity which is typically dependent on nitrate availability. We suggest that changes in light and nitrate availability
initiate a cascade of microbial processes that degrade phytoplankton-based PON into DON, providing a substrate for
ammonia oxidation, and resulting in PNM formation.
**Figure 9. Schematic of nitrite cycling processes and relative DIN pools near the PNM feature. Panel (a) depicts the**
**offshore conditions and panel (b) depicts early upwelling conditions that lead to bloom initiation.**

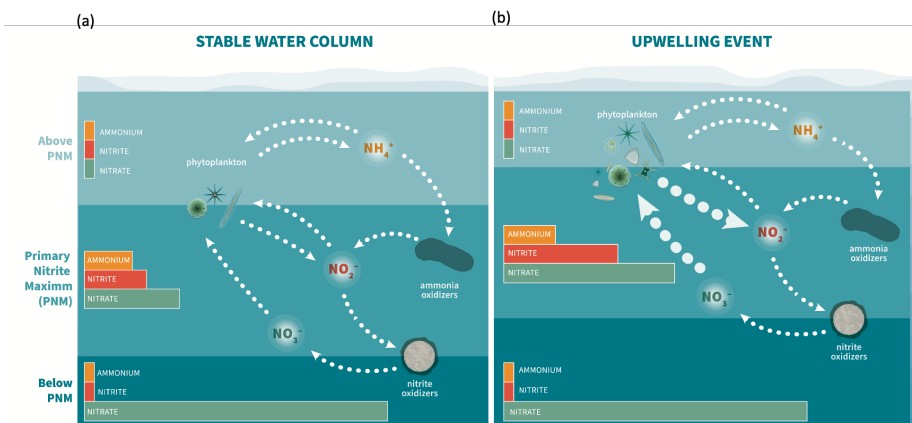






Figure 9 places the findings in the current study in the context of the sequential physical and biological processes
controlling the PNM feature in the ETNP. The schematic depicts a typical offshore PNM observed during a stratified,
stable water column (Fig. 9a), in contrast to that observed during the onset of upwelling (Fig. 9b). In each case, the
surface ocean is split into 3 layers: above, within, and below the PNM, with the PNM sitting near the top of the
nitracline. Phytoplankton control the availability and supply of DIN above the PNM, where high light allows for
complete drawdown of DIN. In the stable water column (Fig. 9a), phytoplankton are present in a chlorophyll maximum
that is small and stable just above the nitracline consisting of smaller eukaryotes and cyanobacteria (Legendre-Fixx,
2017). The chlorophyll maximum is small because there is no active upwelling, and the ambient nitrate at the
chlorophyll maximum has been depleted to low concentrations. Phytoplankton fail to access deeper nitrogen supplies
because light levels become inadequate at depth, so the chlorophyll maximum is balanced at the intersection of the
dual requirements for light and upwardly diffused nitrate. A small ammonium peak develops just below the
chlorophyll maximum, and just above the nitrite maximum, coming from phytoplankton decomposition processes
including grazer activity. The supply of ammonium is adequate to fuel an active nitrifier community in the PNM layer
and below, with average rates of ammonia oxidation and nitrite oxidation near 20 nM day$^{-1}$. The net imbalance in the
two steps of nitrification is small (few nM day$^{-1}$), contributing to the small yet stable accumulation of nitrite at the
PNM. Contributions of nitrite from phytoplankton are minimal because they have drawn down surface nitrate and are
subsisting at the edge of a well-established deep nitracline. Although the water column is stably stratified, the Brunt-
Väisälä values are moderate.
During an upwelling event (Fig. 9b), an influx of nitrate-rich water into the euphotic zone initiates a phytoplankton
bloom. We see evidence of early upwelling at coastal stations where nitrate concentrations at the chlorophyll
maximum are not completely depleted (average 5.2±3.6 μM), while nitrate at offshore station chlorophyll maxima are
lower (average 0.6±0.4 μM). With phytoplankton growth fueled by new nitrate, the ammonium concentration begins
to increase via degradation and grazing, providing substrate for ammonia oxidizers. Rate measurements show a small
increase in average ammonia oxidation rate at coastal stations compared to offshore stations (25.8±3.6 vs. 21.3±3.3
nM day$^{-1}$, respectively). At some coastal stations, a more significant change in the concentration of the nitrite
maximum may come from increased phytoplankton nitrite release. Nitrate reduction rates larger than ammonia
oxidation rates (>30 nM day$^{-1}$) were measured sporadically at stations near the coast. Previous work has documented
up to ~10% of nitrate uptake can be released as nitrite in laboratory culture experiments, suggesting that locations with
high nitrate uptake and active nitrate reduction have the potential for more nitrite release from phytoplankton (Collos,
1998). Additionally, the physical upwelling of deep water compresses density layers in the euphotic zone leading to
higher Brunt-Väisälä values and lower potential for nitrite diffusion away from the site of production, helping to
explain larger nitrite maxima occurring at upwelling sites.
**5 Conclusions**
This study used both high resolution environmental data and direct rate measurements of nitrite cycling processes to
explore the factors contributing to PNM formation in ETNP. At our sites, there was a distinct and predictable depth



where nitrite accumulated in a peak-shaped PNM feature. Linear regression and multivariate regression analysis with
environmental data showed that the top of the nitracline and the top of the oxycline are two major indicators of the
depth of the nitrite maximum. Rate data also showed distinct peaks in activity that corresponded well with the mean
PNM isopycnal for the region. Nitrifier processes dominated nitrite cycling at and below the PNM, while
phytoplankton processes were typically restricted to depths above the PNM. Ammonia oxidation was the dominant
nitrite production process at most depths and stations. We report a handful of high nitrate reduction rates (>20 nM
day$^{-1}$) from coastal stations with higher chlorophyll and nitrate concentrations at the PNM, which suggest there are
opportunities for phytoplankton to play a larger role in nitrite production at the PNM under these conditions. However,
even where nitrite production from phytoplankton remains low, we suggest a sequential and competitive dependence
of ammonia oxidation rates on phytoplankton processes. The importance of co-occurring environmental conditions
and timing lag of microbial interactions should be considered in further work on what determines the formation of
large nitrite maxima. For example, both nitrate and light availability may work together to control net nitrite production
through sequential processes beginning with upwelling events. Microbial physiological responses remain important
in connecting rates of activity to dynamic environmental conditions.
With nitrite production in the PNM predominantly linked to ammonia oxidation, this has potential implications for
production of nitrous oxide in the upper water column of the ETNP. The ETNP is known to be an important source
for atmospheric nitrous oxide (Babbin et al., 2020; Tian et al., 2020), with high accumulations of nitrous oxide in the
near surface (Kelly et al., 2021; Monreal et al., 2022). Nitrous oxide production in the near-surface maximum has
been linked to a combination of hybrid production from AOA, and bacterial denitrification (Kelly et al., 2021; Monreal
et al., 2022; Trimmer et al., 2016). Conditions that favor enhanced ammonia oxidation could thus also promote
enhanced nitrous oxide production and emissions thereby forming a link between stimulation of high primary
productivity and high rates of nitrous oxide production and emission.

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





**Acknowledgements**
The authors acknowledge the captain and crew of the research vessels required to collect this data set:
R/V Ronald Brown, R/V Sikuliaq, R/V Sally Ride and the R/V Falkor. We also acknowledge shipboard
support from Marguerite Blum and Matt Forbes. This research was supported by U.S.-NSF grant OCE-
1657868 to K. L. Casciotti.

**Author Contributions**
Major data collection efforts, data processing/analysis and writing were conducted by N. Travis.
Significant support during data collection was provided by C. Kelly and M. Mulholland, with additional
contributions during manuscript editing. K. Casciotti was instrumental in initial project design, laboratory
analysis, data investigations and manuscript writing.