# Peer review of "Nitrite Cycling in the Primary Nitrite Maxima of the Eastern 2 Tropical North Pacific"

_Biogeosciences, 2022_

## Author Comment (AC1)

**Figures in Reply on RC1**

[Figure]

[Figure]

[Figure]

**Fig. 3.12** Surface currents 1993–2010, March and September climatologies (NASA's ocean surface current analysis-real time project, OSCAR, http://www.oscar.noaa.gov/)

**Fig. 3.16** Annual mean wind-driven upwelling, cm d$^{-1}$, 1950–1997 (Xie and Hsieh 1995). Data provided by Liusen Xie, http://www.ocgy.ubc. a/projects/clim.pred/Upwell/

[Figure]

[Figure]

[Figure]

[Figure]

[Figure]

---

## Author Response (AR1)

**Associate Editor Comment: Reconsider after Major Revisions**

**Comment:** Apologies for the delay in my response - I've been on leave and then at a conference. Many thanks for your contribution to the online discussion and response to the reviewers' comments. Both reviewers were positive overall about the manuscript but provided some constructive comments for improvement. I look forward to receiving your revised manuscript which I will send back to at least one of the reviewers for comment. Please also note the editorial office comment about the colour scheme of your figures.

Thank you for considering reviewer comments and the content of this manuscript. We are appreciative of your time and contribution to improving this work.

**Reviewer 1 Overview:**

**Comment:** The manuscript 'Nitrite Cycling in the Primary Nitrite Maxima of the Eastern Tropical North Pacific' by Travis et al. investigated the distribution and cycling rate of NO2- in the upper ETNP. Statistical analysis and modeling approaches also provide valuable information on understanding the depth and magnitude of PNM. The authors found that the depth of PNM can be well predicted, while the magnitude of NO2- accumulation was less well correlated with any of the measured biological parameters and remained hard to reconcile; instead, several potential reasons were proposed for explaining the varied NO2- maxima.

Overall, I feel the present study is well designed and executed; the main results and findings improve the mechanistic understanding of the formation, distribution, and cycling of NO2- in the upper layer of this global relevant oceanic regime, albeit the reasons responsible for the varied magnitude of NO2- at the PNM remains unresolved. The manuscript is well organized and written despite some parts of the results and discussion appearing to be long and redundant that can be improved.

I have a few concerns and questions regarding the data processing, interpretation, and discussion, and I would like to see the authors' response to these comments (see below).

Thank you for the effort and time spent to improve this manuscript. Major comments and specific comments from Reviewer 1 are addressed below.

**Reviewer 1 Major comments**

**Comment:** The paired light-dark incubation is one of the strengths of the present study. Given that most previous studies used dark incubation for nitrification rate measurement and light incubation to quantify phytoplankton-associated processes rates, the paired light-dark incubation should inform more comprehensive and accurate rates by integrating the rates derived in both conditions. The present manuscript is unclear how the daily rate is derived and whether the results from both light-dark incubations have been incorporated? Meanwhile, comparing the light and dark incubation rates would also help assess the diel rhythm of NO2- cycling in the sunlit ocean.

We thank the reviewer for this suggestion. All the rates used to calculate daily rates in the original manuscript were from incubations at near-in-situ light levels, which in most cases was the low light (LL) tank (~1% PAR). The exception was from shallow depths where the medium light incubation tanks was closer to the ambient light levels at source water collection depths. Each of these in-situ light levels has a paired dark incubation. The daily rates (nM d$^{-1}$) reported in the original manuscript for ammonia oxidation, nitrite oxidation and nitrate reduction were calculated using hourly rates from the ambient light treatments (incubation length of 8 hours starting near dawn) multiplied across 24 hrs, and did not incorporate the dark-incubated rates. Nitrite uptake rates were measured from a 24 hour incubation period that captured a full day:night cycle, and therefore represent a true daily rate.

In response to the reviewer comment, we recalculated the daily rates of ammonia oxidation, nitrite oxidation, and nitrate reduction using the paired dark:light incubated samples where available, applying a simple assumption of a 12 hr:12 hr daily cycle. Dark incubation data were, however, not available for some nitrate reduction measurements in 2016, so those daily rates were calculated using 24 hr light incubated rates. If dark rates of nitrate reduction are lower than the light incubations, these would be an overestimate. Daily nitrite uptake rates were not recalculated due to the 24 hr incubation time mentioned above. These re-calculations had minimal impact on the overall patterns seen in the nitrification rates, because dark and light incubations did not differ significantly. This is an interesting finding in itself, and the average daily rates using 12 hr:12 hr daily cycle vs 24 hr low light daily cycle for ammonia oxidation and nitrite oxidation were 23.0 vs 23.3, and 18.9 vs 19.1 nM/day, respectively. The focus of this study was at depths near the base of the euphotic zone where light inhibition may already be minimal.

- FIGURE TABLE UPDATES: Figure 4 (revised below) and the rate data supplement (Table S1b) have been updated to show the 12hr:12hr daily cycle rates.

- Added [241-243] "Paired dark and light incubation samples were used to convert hourly rates to daily rates using a simple assumption of a 12 hr light:12 hr dark daily cycle."

[Figure]

| Cruise | Depth (m) | Station | NH3.Ox1 (nM d^-1) | Stdev_4 | LOD_4 | NO2.Ox1 (nM d^-1) | Stdev_2 | LOD_2 | NO3.Red1 (nM d^-1) | Stdev_3 | LOD_3 | NO2.up (nM d^-1) | NetNit1 | Net.Nitrif | NetP1 | NetN1 |
|---|---|---|---|---|---|---|---|---|---|---|---|---|---|---|---|---|
| Ward | 60 | PS1 | 7.65 | 0.37 | 0.16 | 6.72 | 4.53 | 0.6 | 0.3 | 0.58 | -0.23 | 5.44 | 0.92 | -3.75 | -5.14 | -4.22 |
| Ward | 55 | PS1 | 13.66 | 3.39 | 0.47 | 4.96 | 3.68 | 1.45 | 5.62 | 1.17 | n.d. | 9.49 | 8.7 | 4.19 | -3.87 | 4.83 |
| Ward | 60 | PS1 | 11.65 | 4.28 | 0.69 | -0.94 | 0.7 | 1.7 | 3.78 | 0.18 | n.d. | 6.04 | 12.59 | 8.73 | -2.26 | 10.33 |
| Ward | 65 | PS1 | 21.59 | 2.24 | 0.95 | 1.9 | n.d. | 2.77 | 3.48 | 0.71 | n.d. | 3.74 | 19.69 | 17.43 | -0.26 | 19.43 |
| Ward | 70 | PS1 | 20.27 | 1.23 | 1.09 | 18.58 | 1.08 | 1.73 | 1.89 | 0.22 | n.d. | 2.01 | 1.69 | 7.9 | -0.11 | 1.57 |
| Ward | 75 | PS2 | 28.78 | 0.46 | 0.16 | 40.32 | 5.43 | 37.18 | 9.14 | 1.26 | n.d. | 8.6 | -11.54 | -25.95 | 0.54 | -11 |
| Ward | 70 | PS2 | -1.2 | 0.11 | 0.12 | -2.62 | 0.52 | 3.26 | n.d. | n.d. | n.d. | 10.43 | 1.43 | 2.07 | n.d. | n.d. |
| Ward | 80 | PS2 | 44.09 | 1.68 | 0.74 | 87.4 | 11.58 | 53.86 | n.d. | n.d. | n.d. | 2.32 | -43.31 | -43.11 | n.d. | n.d. |
| Ward | 30 | PS3 | 90.42 | 1.95 | 0.09 | 60.48 | 2.11 | 0.85 | 5.82 | 0.82 | n.d. | 4 | 29.94 | 33.45 | 1.82 | 31.75 |
| Ward | 30 | PS3 | 68.59 | 2.69 | 1.19 | 42.82 | 1.71 | 0.88 | 1.43 | 0.44 | n.d. | 3.82 | 25.77 | 26.93 | -2.39 | 23.38 |
| Ward | 20 | PS3 | -1.32 | 0.35 | 0.21 | -5.54 | 3.71 | 19.34 | 1.24 | 0.78 | n.d. | 2.28 | 4.22 | -2.7 | -1.04 | 3.18 |
| Ward | 25 | PS3 | 7.09 | 3.2 | 1.12 | 78.56 | 1.6 | 24.95 | 1.58 | 0.22 | n.d. | 1.24 | -71.47 | -49.61 | 0.34 | -71.14 |
| Falkor | 70 | F2 | 57.01 | 34.07 | n.d. | 24.36 | 0.62 | n.d. | n.d. | n.d. | n.d. | 1.06 | 32.64 | 47.94 | n.d. | n.d. |
| Falkor | 65 | F9 | 20.5 | 1.76 | n.d. | 11.1 | 4.98 | n.d. | n.d. | n.d. | n.d. | n.d. | 9.41 | 7.87 | n.d. | n.d. |
| Falkor | 50 | F9 | 1.83 | 0.63 | n.d. | 5.07 | 12.36 | n.d. | n.d. | n.d. | n.d. | n.d. | -3.24 | -9.98 | n.d. | n.d. |
| HODZ | 25 | P1 | 3.63 | 0.06 | 0.04 | -9.2 | 0 | n.d. | 0 | 0 | n.d. | 112.74 | 12.84 | 1.56 | -112.74 | -99.9 |
| HODZ | 30 | P1 | 8.66 | 0.05 | 0.18 | -15.04 | 5.99 | n.d. | 49.13 | 1.02 | n.d. | 165 | 23.7 | 2.08 | -115.88 | -92.18 |
| HODZ | 35 | P1 | 50.45 | 2.92 | 0.95 | -4.09 | 1.1 | n.d. | 33.23 | 5.96 | n.d. | 18.79 | 54.55 | 54.55 | 14.43 | 68.98 |
| HODZ | 45 | P1 | 49.85 | 3.89 | 1.51 | 34.96 | 0.47 | n.d. | 3.54 | 1.6 | n.d. | 4.4 | 14.89 | 14.89 | -0.86 | 14.02 |
| HODZ | 25 | P1 | 4.42 | 1.24 | 0.26 | 29.94 | n.d. | n.d. | 0 | 0 | n.d. | 123.26 | -25.52 | 2.39 | -123.26 | -148.78 |
| HODZ | 30 | P1 | 15.73 | 3.08 | 0.7 | 13.03 | 4.8 | n.d. | 53.15 | 2.15 | n.d. | 49.41 | 2.71 | 8.51 | 3.74 | 6.44 |
| HODZ | 40 | P1 | 26.18 | 4.41 | 1.22 | 45.73 | 4.18 | n.d. | 1.92 | 0.51 | n.d. | 10.83 | -19.55 | -19.55 | -8.91 | -28.46 |
| HODZ | 60 | P1 | 16.87 | 0.09 | 0.98 | 18.35 | 1.92 | n.d. | 0 | 0 | n.d. | 0.66 | -1.48 | -1.48 | -0.66 | -2.14 |
| HODZ | 50 | P2 | -3.15 | 1.05 | 1 | n.d. | n.d. | n.d. | -1.18 | 0.77 | n.d. | 3.99 | n.d. | n.d. | -5.16 | n.d. |
| HODZ | 55 | P2 | 2.11 | 1.08 | 0.4 | 13.41 | 1.91 | n.d. | 2.06 | 0.23 | n.d. | 7.34 | -11.31 | -8.48 | -5.27 | -16.58 |
| HODZ | 65 | P2 | 71.2 | 1.32 | 2.06 | 46.63 | 2.71 | n.d. | 13.12 | 6.29 | n.d. | 6.97 | 24.56 | 24.56 | 6.15 | 30.71 |
| ETNP2016 | 55 | 3 | 0.76 | 0.06 | 0.06 | 8.4 | 0.14 | -0.17 | n.d. | n.d. | n.d. | 46.95 | -7.64 | 0.66 | n.d. | n.d. |
| ETNP2016 | 75 | 3 | 28.17 | 9.34 | 0.83 | 29.25 | 2.12 | 0.99 | -2.3 | n.d. | n.d. | 8.85 | -1.08 | 5.13 | -11.15 | -12.23 |
| ETNP2016 | 100 | 3 | 28.72 | 1.49 | 0.92 | 34.83 | 13.7 | 0.94 | 1.5 | n.d. | 0.19 | 1.07 | -6.11 | -10 | 0.43 | -5.68 |
| ETNP2016 | 120 | 3 | 10.38 | 1.5 | 1.01 | 14.59 | 0.88 | 1.55 | 0 | n.d. | 0 | 1.07 | -4.21 | 1.34 | -1.07 | -5.28 |
| ETNP2016 | 25 | 6 | 2.32 | 0.97 | 0.08 | -8.97 | 27.06 | n.d. | 7.39 | 1.85 | 0.1 | 46.8 | 11.3 | 4.09 | -39.41 | -28.11 |
| ETNP2016 | 40 | 6 | 46.83 | 2.92 | 1.1 | 31.88 | 3.02 | n.d. | 5.56 | 1.08 | -0.1 | 3.5 | 14.95 | 8.75 | 2.07 | 17.01 |
| ETNP2016 | 55 | 6 | 42.59 | 5.33 | 1.67 | 34 | 1.38 | n.d. | 3.07 | 1.56 | -0.08 | 0.69 | 8.59 | 17.42 | 2.38 | 10.97 |
| ETNP2016 | 75 | 6 | 8.36 | 1.89 | 0.81 | 20.98 | 0.56 | n.d. | 2.18 | 0.36 | 0.4 | 0 | -12.62 | -5.66 | 2.18 | -10.45 |
| ETNP2016 | 35 | 9 | 15.42 | 6.28 | 0.44 | 1.71 | 2.37 | n.d. | 0.12 | 0 | 0.01 | 6.64 | 13.71 | 8.35 | -6.52 | 7.19 |
| ETNP2016 | 40 | 9 | 73.84 | 1.12 | 1.16 | 5.48 | 0.43 | n.d. | 12.33 | 1.9 | 1.33 | 7.18 | 68.37 | 30.34 | 5.15 | 73.52 |
| ETNP2016 | 50 | 9 | 59.28 | 16.31 | 1.62 | 20.89 | 0.2 | n.d. | 0.41 | 0.49 | 0 | 3.35 | 38.39 | 33.16 | -2.94 | 35.45 |
| ETNP2016 | 60 | 9 | 13.63 | 4.46 | 1.93 | 7.54 | 1.53 | n.d. | 0.04 | 0 | 0.01 | 0.54 | 6.09 | 6.75 | -0.5 | 5.59 |
| ETNP2016 | 10 | 12 | -0.42 | 0.1 | 0.05 | n.d. | n.d. | n.d. | n.d. | n.d. | n.d. | n.d. | n.d. | -0.42 | n.d. | n.d. |
| ETNP2016 | 75 | 12 | 2.85 | 0.4 | 0.06 | 25.64 | 6.93 | n.d. | 0.9 | 0.22 | -0.01 | 19.04 | -22.79 | -21.63 | -18.14 | -40.93 |
| ETNP2016 | 85 | 12 | 0.04 | 0.04 | -0.03 | 0.37 | 37.79 | n.d. | 0.04 | 0.01 | 0 | 17.4 | -0.33 | -0.06 | -17.36 | -17.7 |
| ETNP2016 | 95 | 12 | 30.19 | 0.09 | 0.26 | n.d. | n.d. | n.d. | 0.06 | 0.04 | 0.04 | 21.26 | n.d. | n.d. | -21.2 | n.d. |
| ETNP2016 | 100 | 12 | 32.15 | 0.76 | 0.48 | 5.8 | 0.08 | n.d. | 0.78 | 0.3 | 0.63 | 11.18 | 26.36 | 27.09 | -10.41 | 15.95 |
| ETNP2016 | 62 | 16 | 0.55 | 0.16 | 0 | n.d. | n.d. | n.d. | n.d. | n.d. | n.d. | n.d. | n.d. | n.d. | n.d. | n.d. |
| ETNP2016 | 72 | 16 | 31.52 | 3 | 0 | n.d. | n.d. | n.d. | n.d. | n.d. | n.d. | n.d. | n.d. | n.d. | n.d. | n.d. |
| ETNP2016 | 80 | 16 | 53.57 | 3.4 | 0 | n.d. | n.d. | n.d. | n.d. | n.d. | n.d. | n.d. | n.d. | n.d. | n.d. | n.d. |

**Comment:** I have concerns about using the gross rates derived from the high tracer enrichment (i.e., 200 nM). Because the NO2- concentration drops sharply outside the PNM, and the NH4+ concentration appears to be low in most of your incubation depths (Table S1), the rates reported here should be attributed to 'potential rates.' While I acknowledge these potential rates are still very valuable, care should be taken in interpreting these results. For example, the authors measured some conspicuous high rates under substrate depleted samples, such as those high NO2- uptake rates (> 100nM/d) above the PNM where ambient NO2- is low. It tells the high potential for the phytoplankton to control the cycling and distribution of NO2-, but is it meaningful to use those potential rates to calculate the residence time?

We agree that the measurements presented are best characterized as *potential rates* and are likely stimulated by the addition of a large, but consistent, amount of [15]N tracer. We have noted this caveat in the manuscript (lines 183-185). To investigate this further, we have added plots of substrate vs.

nitrification and nitrite uptake into the Supplement (lines 166-170, also see figure below). We acknowledge that many high rates occurred at the lower end of the range of ambient substrate concentrations, where the addition of 200 nM $^{15}$N tracer may indeed represent a significant alteration. However, not all low DIN samples resulted in high rates. This suggests that the potential rates were not uniformly increased by the addition of additional substrate, and DIN is not always rate limiting. Other conditions of the source water at our sampling depths (with potentially variable abundances of active microbes) likely also played a role in the observed variation in rates at low substrate concentrations.

[Figure]

While we cannot conclusively show whether low per volume rates were a result of low cell concentrations or lack of enhancement from tracer addition, the spatial distribution of rates plotted in Figure 4 shows how samples collected from a variety of ambient conditions – and therefore experiencing varying levels of rate enhancement due to $^{15}$N additions and varying cell concentrations – still reveal vertical zonation in maximum activity for each rate process. It remains unclear what controls this vertical distribution, although the MLR analysis suggests that nitrate and light are important environmental parameters relating to nitrite accumulation. Experiments that directly test environmental controls on microbial rates are needed, such as light and DIN manipulation experiments. While the rates presented here are per liter, not per cell, they are still useful in understanding bulk water column nitrite transformation.

With regards to the residence time estimates, we agree with the reviewer that our estimates of average residence time could be underestimated because of potential enhancement from tracer additions, and this additional caveat is now noted in the revised manuscript (line 464-465). We also note that steady state dynamics are an additional assumption, and unlikely to be true in all cases (lines 458 and 471). Residence times were on the order of days to weeks (albeit with a very large range). Perhaps summarizing these residence times into a mean was an oversimplification that hides the variance displayed in Fig S6, especially above the nitrite maxima. The median residence time (using total production input) was 7.8 days (line 460), while the mean was 30.4 days. We are also likely missing an input/output term from physical mixing, which could have a larger influence in dynamic coastal waters compared to offshore. We have mentioned this in the revised manuscript (line 470). Separating coastal stations and offshore stations shows that residence time estimates from rate data alone were slightly longer at offshore stations. Median data have been added to the manuscript (line 468).

        Coastal - mean 17 days, median 5.8 days

        Offshore  - mean 53 days, median 18.2 days

It is valuable to consider the differences in potential residence times between coastal and offshore conditions, even with the limitations and assumptions of making residence time estimates from potential rates, where the rates may be enhanced by tracer addition. It is likely that multiple methods will be needed to more accurately constrain the residence time of nitrite in the ETNP PNM, and this is suggested in the manuscript (line 474).

- Added [183-185] "High tracer enrichment in samples with low ambient concentrations may lead to enhancement of rates, which are best characterized as potential rates; care must be taken when interpreting results ."

- Added [239-240] "While the initial 15N in the substrate pool was directly measured at time zero for use in rate calculations, the addition of 200 nM 15N tracer led to variable atom fraction of 15N in the substrate pool across experiments, which was calculated from ambient and tracer substrate concentrations."

- Added [468-473] "Our estimates of average residence time using potential rates may be underestimated because of rate enhancement from tracer additions, and we are also likely missing an input/output term from physical mixing, which could have a larger influence in dynamic coastal waters compared to offshore. Comparing coastal and offshore stations, the estimated residence times are quite different between regimes (mean residence times of 17 and 53 days, respectively, and median residence times of 5.8 and 18.2 days, respectively) suggesting that coastal nitrite accumulations are turning over more quickly even with the limitations and assumptions of these calculations. ."

- Added [464] "…while the median residence time was 7.8 days"

- Added [477-478] "Additional methods of estimating nitrite turnover rate, such as using variation in natural abundance nitrite isotopes, may provide more insight (Buchwald and Casciotti, 2013)."

**Comment:** The prominent accumulation of NO2- (i.e., >1µM) in the coastal stations is not surprising. It is interesting to see the absence of higher ammonia oxidation rates in these more eutrophic systems, as is frequently observed in other studies. On the other hand, the authors observed a mild increase of NO2- released by the phytoplankton, but I expect a long time is still required to get the high NO2- concentration observed here. The question is, can you find evidence of such a long residence time of the water mass here, as I expect a short residence time of these shallow, high dynamic coastal waters?

Coastal waters are, indeed, typically more dynamic than offshore, and the ETNP experiences upwelling along the coast. Increased wind-driven surface currents are expected on the southern coastline of our study area in the spring months (see cropped figures Fiedler and Lavín, 2017, below).

[Figure]

**Fig. 3.12** Surface currents 1993–2010, March and September climatologies (NASA's ocean surface current analysis-real time project, OSCAR, http://www.oscar.noaa.gov/)

**Fig. 3.16** Annual mean wind-driven upwelling, cm d⁻¹, 1950–1997 (Xie and Hsieh 1995). Data provided by Liusen Xie, http://www.ocgy.ubc.ca/projects/clim.pred/Upwell/

The surface current data from April 07-11, 2016 (5-day average from OSCAR; Earth & Space Research, below, added to Supplement Fig S8) showed fastest movement of surface waters near the southern coastal stations (6, 7, 8, 9), similar to the averaged March surface currents by Fiedler and Lavín (2017). This subset of 4 southern coastal stations (used to inform the coastal MLR) have the largest observed nitrite maxima (800-1400 nM), larger chlorophyll maxima and shallower nitraclines. Rates were measured at stations 6 and 9 of this subset, where ammonia oxidation reached 73.8 and 46.8 nM/day, respectively. At the 40 m depth at station 6, low rates of nitrite uptake and nitrite oxidation, led to relatively a high net nitrite production rate of 73.5 nM/day (with total production at 86 nM/day). Considering this net production rate and the observed 824 nM nitrite maximum, the nitrite accumulation would take a number of days to accumulate. However, even with the faster surface velocities at these stations, a median residence time for nitrite around 6 days seems feasible.

- Figure S8 added to Supplement [172-175]

[Figure]

In addition, the density gradients at the coastal stations and the calculated Brunt-Vaisala frequencies give some insight into physical mixing of water near the PNM, as high BV values indicate strong density gradients and reduced vertical mixing. The highest BV values (strongest density gradients) were found at the coastal stations, and corresponded with some of the largest nitrite maxima (Fig 2I, and station density profiles below). The full density profiles have been added to the supplement (Fig S9). As discussed in the manuscript, nitrite produced at coastal stations within compressed density gradients may not be diffusing/advecting vertically away from the production depth, thus leading to higher concentrations at those nitrite maxima. Our interpretation is that the higher rates together with the increased vertical stability allow the coastal stations to accumulate more PNM than offshore stations, and while the water moves through the coastal zone relatively quickly, its residence time in the coastal zone is long enough to support the required accumulation of nitrite over its 6 day residence time. This the following was added to the revised manuscript (lines 740-747):

- Edited [Line 743-747] "The four southern coastal stations (used to inform the coastal MLR) had the largest nitrite maxima measured in this study (with nitrite concentrations reaching 800-1400 nM). However, it is reasonable to expect that in dynamic coastal waters, upwelling and offshore transport of water would lead to shorter water residence times and less time for nitrite to accumulate in the PNM. Indeed, local surface current data from early April 2016 show the fastest currents occurring along the southern coastline (Fig S8). However, even given these current velocities, nitrite accumulation over the span of days to weeks seems possible. Thus, our nitrite residence time calculations, on the order of days to months, are consistent with the residence time of water in the coastal environment, and other estimates of PNM residence times (Fig. S6). "

- Figure S9 added to Supplement [187-19

[Figure]

**Comment:** I am also very interested to know if the main findings (both the contributing processes and time required for PNM formation) derived from the tracer experiment are consistent with dual NO2-isotope (i.e., 15N-18O natural abundance) in this region (if any). To my knowledge, the combined use of the natural abundance and isotope labeling approaches in exploring the PNM remains lacking, so any comparisons between these two independent methods would be very helpful.

We fully agree that using multiple approaches to understanding the PNM formation would be informative, and we plan to contribute that piece in a subsequent paper. However, the analysis and interpretation of the natural abundance nitrite isotopes is a substantial undertaking in its own right, and it is our assessment that adding it to the current paper would be unwieldy. Hopefully future analysis of the corresponding dual nitrite isotopes from this cruise will offer an opportunity to compare rates estimates (and residence time estimates) between these two methods.

- Added [477-478] Additional methods of estimating nitrite turnover rate, such as using variation in natural abundance nitrite isotopes, may provide more insight (Buchwald and Casciotti, 2013).

**Reviewer 1 Specific Comments**

**Line 41:** Not sure how the authors derive the concentration of 300nM? Is it based on any specific statistics? As NO2- concentration at the PNM varied over space and time (as highlighted in the manuscript) and frequently fell below 300nM (e.g., in the subtropical gyres), justification is needed to better clarify the number.

The ~300 nM average PNM nitrite concentration was derived from the GLODAP Pacific Ocean nitrite measurements. Cast data from the Pacific Ocean was filtered to remove data with <3 µM oxygen or any data below 400m to exclude any secondary nitrite maximum measurements from oxygen deficient zones. PNM size was calculated from the maximum nitrite concentration at each station. This analysis yielded maxima ranging from 0 to 3.29 µM (excluding a single value >10µM), with an overall average primary nitrite maximum of 252 nM across 3485 stations. Because mean values can be skewed by outliers, we also calculated the median PNM size across the GLODAP Pacific dataset, median nitrite maximum of 237 nM. These values have been clarified and revised in the manuscript (line 40).

- Edited [40-42] "In the Pacific Ocean, the median nitrite concentration across PNM features is 237 nM (GLODAP), although concentrations as high as 2.8 µM have been reported (Brandhorst, 1958; Carlucci et al., 1970; Dore and Karl, 1996; Wada and Hattori, 1972; GLODAP, V2)."

**Line 56:** I suggest revising the description of the three microbial groups to ammonia oxidizers, nitrite oxidizers, and phytoplankton.

Yes, microbial groups will be clarified as requested in the revised manuscript (line 55).

- Edited [53-54] "Near the PNM, the three main microbial groups involved in nitrite cycling are ammonia oxidizers, nitrite oxidizers and phytoplankton."

**Line 57:** Bacterial ammonia oxidizers should not be excluded as they might also play a role in some regions, such as the coastal zone.

We agree that bacterial ammonia oxidizers should not be excluded, and our rates represent the sum of all ammonia oxidizers in the community, since they were measured as bulk community rates. We did not attempt to identify what components of the ammonia oxidizing community were most active. Our intention was simply to note that archaeal ammonia oxidizers are typically dominant.

**Lines 138-154:** Please clarify the detection limit and accuracy of the methods used for on-board and onshore nutrients measurement.

Detection limit for shipboard spectrophotometric nitrite measurements is typically 200 nM (Strickland and Parsons, 1972) and in general agreement with what we directly derived from our standard curves. Detection limits were added to the manuscript.

- Edited [133] "…with a detection limit of ~200 nM"
- Edited [144] "...and detection limit of 85 nM (Miller et al 1998, Rajacovic 2008)."

**Line 151:** The acronym of PPS has been defined in line 136.

The PPS acronym definition has been removed from line 146.

**Lines 180-181:** I have concerns about using the HDPE Nalgene bottles for light incubation due to the screen of light by the HDPE bottle. PC bottles are more widely used for simulated light incubation. Has the light intensity been measured inside the bottles? How accurate was the actual light intensity compared to the in-situ light intensity?

We apologize for the error in the original manuscript. The light incubations were indeed all conducted in clear polycarbonate bottles, as the reviewer states is typical for light incubations. It has been corrected in line 199. The brown bottles used for dark incubations were HDPE, and we made a typographical error in describing the light incubations in the original manuscript. The light levels were not measured inside the PC incubations bottles, but each incubation tank was tested using a PAR meter to determine the incubation light levels. Percent surface PAR at each collection depth was also measured (see Table S1a).

- Edited [179-181] "From each depth, six clear 500-ml polycarbonate (PC) Nalgene bottles were triple-rinsed and filled directly from the Niskin bottle for light incubations. Additionally, six 500-ml or 1-L amber high-density polyethylene (HDPE) Nalgene bottles were triple-rinsed and filled for paired dark incubations."

**Line 213:** For the samples with low in-situ $NO_2^-$ concentration and high $NO_2^-$ consumption rate (either by $NO_2^-$ oxidation or assimilation), the newly produced $^{15}NO_2^-$ via $^{15}NO_3^-$ reduction might have been utilized by NOB or phytoplankton. The observed $NO_3^-$ reduction rate thus might represent a conservative estimate.

Yes, we agree it is possible that estimates of nitrate reduction were confounded by uptake of the $^{15}N$ labeled product. We have noted this issue in the manuscript (line 706 - 710). In response to this comment, we explored our data for any patterns that might reveal the described issue. In reference to the figure below, we found that low nitrate reduction rates were found throughout the range of nitrite consumption rates. In addition, we found high nitrate reduction rates even at high rates of nitrite consumption. Two points showing low nitrate reduction rates at high rates of nitrite consumption were investigated further. These points derived from a coastal station (PS1) during the winter 2017 cruise at the shallowest depths measured (25 m), where nitrite concentrations were near zero. It is possible that nitrate reduction measurements are underestimated for these samples.

[Figure]

- Added [709-713] "Because our measurements were of whole community rates, a variety of microbial processes may have remained active in the incubations alongside the process intended to be traced with $^{15}$N. For example, the 15N-NO2- produced via nitrate reduction is potentially acted upon by nitrite uptake and nitrite oxidation. This has the potential of leading to underestimation of nitrate reduction rates, especially where nitrite concentrations are low, and nitrite uptake and nitrite oxidation rates are large."

**Line 233:** I have two questions/ concerns regarding the rate calculation: 1) the equation is used to derive the gross (i.e., in-situ NH4+ plus 15NH4+) oxidation rate (potential rate), taking the different f15N for samples from various depths and stations, the enhancement of the rate due to tracer enrichment would also be varied. 2) it is unclear to me how the daily rate was calculated, have the rates from light and dark incubation been considered? These issues should be clarified.

1) We thank the reviewer for their comments. It's true that we did not correct for variability in spike enhancement of the potential rate (as noted in line 267). This issue would mainly affect $NH_4^+$ and $NO_2^-$ -dependent rates because ambient DIN is highly variable near PNM depths. This is part of the reason why a single concentration was selected for $^{15}$N tracer addition, since it quickly became unfeasible to measure all ambient DIN pools and calculate a 10% tracer addition for each experimental sample. I believe this concern will have to be addressed as a caveat to the dataset, stating that these are *potential rates* with a high likelihood of being stimulated by the tracer additions in most incubations, and that the relative enhancement of the rates may vary across depths due to different ambient DIN concentrations (see line 183 and 239). However, the initial $^{15}$N fraction in the experimental substrate pool was still calculated using the ambient concentrations and the tracer addition in order to determine final rates. Another limitation when comparing rate measurements across depth, is that these are bulk rates per volume, not specific per cell rates, and it is not known how cell concentrations varied with depth in these samples.

- Added [183-185] "High tracer enrichment in samples with low ambient concentrations may lead to enhancement of rates, which are best characterized as potential rates; care must be taken when interpreting results."

- Added [239-241] "While the initial 15N in the substrate pool was directly measured at time zero for use in rate calculations, the addition of 200 nM 15N tracer led to variable atom fraction of 15N in the substrate pool across experiments, which was calculated from ambient and tracer substrate concentrations."

2) Daily rates were originally calculated using only the light incubation rates, converted to an hourly rate and scaled to 24 hr. In response to the reviewer comment, these rates have been recalculated based on a 12 hr:12 hr daily cycle using light incubation rates for the 12 hr light and dark incubation rates for the 12 hr dark, and the results were updated in Figure 4 and Table S1. See also Major Comment 1.

**Lines 311-312:** The criteria for defining coastal and offshore stations should be clarified, e.g., by using the isobath or distance to the shore? Mark the boundary of the 'coastal zone' in Fig. 1a should also be helpful.

The construction of the multiple linear regression models used subsets of example stations from 2016 (four each) to train the "coastal" and "offshore" models, although there are other stations within the complete data set that could be characterized as coastal or offshore. Although they are not strictly defined spatially, we used the wording "coastal" because those water column traits are more common at coastal stations experiencing upwelling. The coastal stations were selected based on water column traits such as their larger chl maximum, shallow nitracline, and shallow mixed layer depth using pump profiler cast data. Offshore stations were selected based on distance away from the coast.

In the revised manuscript, we updated the map by encircling all the coastal stations in green to help visualize where data were drawn for the coastal stations (see updated map below). We have also explained in the MLR section that only a subset of 4 "coastal" stations from 2016 (Stations 6,7,8,9) were selected because they had water column features that were most representative of upwelling-like conditions (eg. shallow nitracline, larger chl maxima).

- Edited [282-287] "Subsets of stations were selected as exemplary of the coastal and offshore regimes based on proximity to the coast, concentration of the chlorophyll maxima, and nitracline depths. The selection criteria for coastal stations used in MLR construction included being close to a coastline, nitracline <40 m depth and chlorophyll maximum larger than 9.5 mg m-3. Offshore stations were selected based on furthest distance from a coastline. Not all stations proximal to the coastline were characterized as coastal (see map) nor included in the 'coastal' subset used to train the model (Table S2a, b)."

- Map EDITED [line 335]

[Figure]

**Line 349 and line 373:** Fig. 2 and 3 contain a lot of statistical information that is very useful. But even though the regression of NO2- maxima /PNM depth against the parameters (listed in Table 1) was not fully shown in Fig. 2 and 3. Would you consider plotting a heatmap showing the result of Pearson correlation analysis (or other statistical information) between NO2- maxima /PNM depth and the collected parameters? That would provide more comprehensive information and save some space at the same time.

Yes, we agree that this would be helpful. The correlations have been added in Table S2c.

- Table S2c added to Supplement [line 21]

| | Pearsons R | | | P-value | |
| | PNM_max | PNM_depth | | PNM_max | PNM_depth |
|---|---|---|---|---|---|
| PNM_max | 1.000 | -0.521 | PNM_max | NA | 0.03871 |
| PNM_depth | -0.521 | 1.000 | PNM_depth | 0.03871 | NA |
| Chl_max | 0.527 | -0.577 | Chl_max | 0.03599 | 0.01923 |
| NH4_max | 0.312 | -0.494 | NH4_max | 0.23958 | 0.05205 |
| Chl_depth | -0.624 | 0.836 | Chl_depth | 0.00985 | 0.00005 |
| NH4_depth | -0.505 | 0.787 | NH4_depth | 0.04612 | 0.00030 |
| Nitracline_top | -0.549 | 0.974 | Nitracline_top | 0.02771 | 0.00000 |
| Nitracline | -0.583 | 0.904 | Nitracline | 0.01773 | 0.00000 |
| Oxycline_top | -0.575 | 0.868 | Oxycline_top | 0.01978 | 0.00001 |
| Oxycline | -0.520 | 0.716 | Oxycline | 0.05649 | 0.00398 |
| pPAR1 | -0.593 | 0.931 | pPAR1 | 0.02555 | 0.00000 |
| ChlPNM | 0.502 | -0.478 | ChlPNM | 0.04732 | 0.06117 |
| NH4PNM | -0.055 | -0.411 | NH4PNM | 0.83873 | 0.11371 |
| NitPNM | 0.733 | -0.623 | NitPNM | 0.00122 | 0.00994 |
| TempPNM | 0.552 | -0.446 | TempPNM | 0.02669 | 0.08314 |
| PNM_sig | -0.484 | 0.395 | PNM_sig | 0.05762 | 0.13024 |
| PARPNM | 0.253 | -0.680 | PARPNM | 0.34348 | 0.00372 |
| OxyPNM | -0.578 | 0.001 | OxyPNM | 0.01907 | 0.99595 |
| BV | 0.664 | -0.183 | BV | 0.00507 | 0.49852 |
| Int_Chl | 0.532 | -0.516 | Int_Chl | 0.03379 | 0.04074 |
| Int_NO3 | 0.698 | -0.941 | Int_NO3 | 0.00263 | 0.00000 |
| Int_NO2 | 0.853 | -0.604 | Int_NO2 | 0.00003 | 0.01325 |
| Int_NH4 | -0.337 | -0.114 | Int_NH4 | 0.20223 | 0.67487 |

In addition, as you have measured NO2- uptake rate, I assume you have also measured the PN concentration, which is a better indicator of organic matter stock and biological productivity than the Chl-a; thus should be included in the main results, and the statistical analysis.

Yes, we do have some PN concentration data, but they are limited to stations and depths where an uptake rate was measured, so there would be fewer data to use in the regression analyses if based on PN concentration data. Therefore, we have chosen to retain the correlations to Chl-a, as a proxy for PN concentration.

**Lines 434-435:** These results suggest either a quasi-homeostatic or physical dispersion plays a role in determining the magnitude of PNM.

Yes, we investigated this through the Brunt Vaisala frequencies, which correlated with PNM size. These correlations suggest that interaction with physical processes may indeed play an important role in formation and maintenance of the PNM feature. As we discuss, the concentration at the nitrite maxima cannot be explained with just the rate measurements. We investigated this further in response to Major Comments (above).

**Section 3.6 and 3.7:** These sections are long and read dense; many results are similar between the sections. There are also a lot of similar descriptions in section 4.3. I suggest reducing these parts and focusing on the main findings.

Thank you for this suggestion. In the revised manuscript, all MLR sections have been condensed and details moved to the supplement.

[revised manuscript text omitted]

**Lines 609-610:** I concur that a net production rate is required for the formation and maintenance of NO2- accumulation at the PNM. On the other hand, negative feedback, either physical dispersion and/ or enhanced biological NO2- consumption, must be involved to restrict the magnitude of PNM to a certain degree. That saying, NO2- accumulation at PNM should be in the quasi-steady state that varies over the diel cycle, seasonal cycle, or some event disruption, making the concentration of NO2- at PNM less predictable.

 This is a valuable insight, and we agree that the PNM is an observable manifestation of a lot of moving variables. And while the feature is ubiquitous, and seems constrained in terms of depth (always the base of the euphotic zone), its size (although usually in the 100-1500 nM range) is not easily predictable by snapshot measurements in the water column, such as environmental data or rate measurements because of periodically changing environmental conditions. Mackey et al. (2011) investigated the formation of a PNM at a single location over time, and the PNM feature grew from low concentration (vertically homogenous) to a peak-shaped profile reaching near 1 uM at its peak in just 7 days with the onset of stratification and a chlorophyll bloom. It would be interesting to see how rate measurements might change along a sequence of 7 days during PNM formation. We have tried to capture this interplay between physical and biological controls in the revised manuscript.

**Line 650:** See my comment on sections 3.6 and 3.7 above. Please consider reducing these sections to avoid redundancy.

Thank you for this suggestion. All MLR sections have been highly reduced/summarized and details moved to the supplement in the revised manuscript.

**Lines 710-711:** Check the format.

Thank you, we have removed the underline format.

**Lines 749-751:** Again, these results indicate a dynamic feature of NO2- concentration at the PNM, which can't be fully explained by a snapshot of the measured rate.

We fully agree that while rate measurements are useful, more information may be needed in order to predict nitrite profiles accurately. Modeling efforts beyond simple MLR analysis may be a good way to combine information gained from direct rate measurements with natural abundance isotope constraints and physical processes over longer time scales.

- Added [732-735] Modeling efforts that are able to integrate both physical diffusion of nitrite and mixing around the PNM, as well as the influence of environmental fluctuations on microbial rates over longer time scales may be more able to explain observed nitrite concentrations. Additional data from time-integrated approaches such as natural abundance nitrite isotopes would also contribute to estimating nitrite age in the PNM.  .

**Line 829:** It looks to be the only difference between panels a and b is the role of phytoplankton in releasing $NO_2^-$, with a higher rate in the more eutrophic coastal upwelling zone. At the same time, the role of nitrifiers remains unchanged. The question is: what is the potential cause for the absence of stimulation on nitrification rate to the enhanced productivity (and thus organic regeneration and substrate supply)?

The schematic in Figure 9 was built using the observations from this dataset, namely that mean nitrification rates were not significantly different between the offshore stations and the coastal stations, while all of the highest nitrate reduction rates were coastal (see replotted Figure 4 below). The mean rates of ammonia oxidation were not significantly different between coastal stations (28.0 nM/day) and offshore stations (20.3 nM/day), t(44)=1.091, p=0.28. Rates of nitrite oxidation were also not statistically different between coastal (20.4 nM/day) and offshore stations (18.0 nM/day), t(28)=0.318, p=0.752. However, there was high variance in the rate measurements, and all some of the highest rates of nitrification were found at coastal stations (eg. 2018 PS3 and 2016 Station 9). Table of mean values was added to the supplement (line 11-12).

We agree that it makes sense that enhanced productivity, regeneration, and nutrient cycling would lead to higher rates of nitrification. We could speculate on why this was not observed, such as the absence of an active nitrifying population, inhibition by light, or competition with phytoplankton for the released ammonium. To answer this question, we would likely need additional information, such as cell counts, or experimental data addressing light inhibition. Another question is whether the depth-integrated rates of ammonia oxidation are higher at coastal stations. We do have a separate dataset on experimental manipulations that will be analyzed for a subsequent manuscript. Unfortunately, meaningful depth integration of rates from 3-4 source water depths per profile proved difficult and highly uncertain.

[Figure]

- Table S1c added to Supplement [11-12]

| Rates | Median | | Mean | | t-value | p-value |
|---|---|---|---|---|---|---|
| | Coastal | Offshore | Coastal | Offshore | | |
| Ammonia Oxidation | 16.3 | 16.97 | 28.0 | 20.3 | 1.093 | 0.281 |
| Nitrite Oxidation | 19.62 | 8.91 | 20.4 | 18.0 | 0.318 | 0.752 |
| Nitrate Reduction | 1.58 | 1.89 | 7.88 | 3.08 | 1.410 | 0.170 |
| Nitrite Uptake | 4.2 | 7.15 | 25.96 | 8.52 | 1.874 | 0.073 |
| Net Nitrite Production | 3.18 | 1.57 | -8.99 | -0.69 | -0.667 | 0.510 |

**Lines 859-861:** On the other hand, won't the upwelling reduce the residence time of the water mass at PNM due to the replacement of water by the upwelling? Also, can you see any clues of NO2- supply from SNM as I expect significantly higher NO2- can be seen in the oxygen-depleted subsurface water in at least part of your stations?

The SNM and PNM are separated spatially in the water column, so that SNM nitrite likely doesn't overlap with the PNM at these stations. All data associated with samples containing <3µM oxygen were removed from analysis. It's an interesting question about the effect of upwelling on the residence time of water in the PNM. Unfortunately, we don't have age tracers together with our data and would be speculating on the residence time of the water itself. The short residence time of the nitrite in coastal waters (around 6 days) does seem to be in keeping with a dynamic coastal environment, which we now address in the revised manuscript

**Supplementary Line 70:** What does this panel show?

Figure S7 showed a comparison of the observed nitrite concentrations and modeled nitrite concentrations from the full-variable multiple linear regression analyses. A bottom panel showed how the "coastal" MLR model (built using 4 example coastal stations) has a poor fit when applied across all 16 stations. These figures have been removed from the final supplement.

**Reviewer 2 Overview**

This study by Travis et al., entitled 'Nitrite Cycling in the Primary Nitrite Maxima of the Eastern Tropical North Pacific' investigates the roles of four major processes that affect the depth and maximum concentration of $NO_2^-$ in the primary nitrite maximum (PNM). A suite of experimental and modelling techniques are applied for several cruises in the ETNP, and show that the depth of the PNM is correlated with water column parameters such as the chlorophyll, oxycline and nitricline depths, while the concentration of $NO_2^-$ at the PNM is weakly correlated. This study confirms many prior studies in other ocean locations and adds to the field by addressing characteristics of coastal/upwelling PNM formation.

Overall the manuscript is thorough and uses a robust combination of approaches to tease out oceanographic processes that affect PNM formation in the ETNP. The study is also well-framed and the literature review is used to give a clear context to the results. The manuscript becomes redundant at times, and the authors might be able to streamline these parts in the interest of space.

We appreciate the constructive comments and agree with the need to streamline parts of the manuscript. We have especially focused on shortening the multiple linear regression modeling sections and moving details to the supplement.

**Comment:** I found the coastal/oceanic comparison interesting and novel, but was unsettled by the lack of clear defining features used to classify each site. The authors state that coastal sites were selected based on "presence of shallow nitraclines and shallow chlorophyll maxima depths, as well as larger chlorophyll maxima and nitrite maxima. […and] had the steepest density gradients near the observed larger PNM." While these criteria are logical, their major weakness is that they rely on the measured data in order to group the sites, and then the same sites are modeled against the same data set, making it somewhat circular. The sites should be grouped based on other criteria that are independent of the measured parameters (e.g. isobath or distance from shore).

In the first version of our multiple linear regression model we included all the station data possible and assessed the best fit model for our entire ETNP dataset (full variable - all station model, Fig 6). This first modeling effort was unable to accurately predict nitrite profiles for all the stations, with some of the largest size errors occurring in nitrite maxima occurring at Stations 5-9, effectively the coastal stations (Table S5). We hypothesized that the controls on nitrite accumulation in the PNM might be different across stations and between regions within the ETNP, especially for stations within a productive upwelling coastal zone.
Our coastal vs offshore categories were a result of inspecting the CTD station data and noticing patterns in the hydrography and chemical gradients of stations with the largest nitrite maxima. Using the measured data we refined the spatially-defined coastal stations into a smaller 4-station subset based on the size of the nitrite maximum, its depth, the size of the chlorophyll maximum and the depth of the nitracline. The selection criteria for "coastal" stations used in MLR construction were: near the coast, top of the PNM feature beginning <40m depth, and chlorophyll maximum larger than 9.5 mg m$^{-3}$. To build our models, we employed a k-fold cross validation method to avoid having to withhold large subsets of data for training and validation prior to testing the model against a small remaining set of field observations. In this way we have avoided the common modeling pitfall of predicting data that were used to build the model in the first place.

A general finding across all models was that the depth errors were small, while size of the nitrite maximum was not as accurately predicted. We believe that predicting nitrite profiles at the coastal stations using the "coastal" model is acceptable due to the k-fold cross validation methods used, and we did find that both regional models performed better than 'all station' models. However, the goal of this modeling effort was not simply to predict nitrite. We aimed to compare the resulting coefficients from a model that reflected the environmental conditions associated with larger nitrite peaks near the coast vs the conditions of an offshore station (Discussion 4.3). This comparison helped to investigate the underlying controls on the accumulation of nitrite at the different station groupings. From the relative importance calculations for each coefficient, we discerned that the coastal model nitrite predictions were influenced by nitrate and light, while the offshore model nitrite predictions were still dependent on nitrate concentrations but had more influence from chlorophyll concentration. Thus, we believe the finding that nitrite at the 'coastal' and 'offshore' stations has a different relationship to the environmental parameters is a distinct and valid finding from this analysis.

**Lines 515-519:** The ability of the model to predict the formation of double PNM peaks is intriguing – the model predicted the feature at 5 sites, of which 3 showed the double peak and two did not in the field data. The manuscript would really benefit from some discussion as to the possible disconnect observed here because it could elucidate important timing or hydrographic factors that influence PNM formation. For example, do the authors believe the double peak is due to $NO_2^-$ formation/consumption rates changing rapidly with depth such that the feature is too transient to consistently observe in field profiles? Is there any evidence to suggest whether the smaller double peak is a remnant of a prior peak that is degrading, or a new peak that is "growing in", (or both)? Do the authors think this is due to physical factors, like shoaling or mixing, or chemical factors that influence the biota, such as upwelling?

Thank you for your comments. The double-peaked PNM is an interesting occurrence that is not well studied, likely because it is less commonly observed. The high resolution PPS nitrite profiles allowed us to observe small additional nitrite peaks at 4 of the 16 stations occupied in 2016 - two on the underslope side and two on the upper slope side of the main PNM. Prediction of additional nitrite peaks by the MLR models occurred fairly frequently, but usually erroneously. However, the predicted double peaked nitrite profiles suggest that the environmental parameters included in the model contain enough variation, or interactions across depth, to introduce these additional peaks in the nitrite profiles.
The reviewer raises several interesting questions about the connections between microbial rates, environmental conditions and observed nitrite accumulations varying over time scales not well captured by our current dataset. Our measured rates of microbial processes may only reflect instantaneous changes in nitrite, which may be disconnected from absolute nitrite concentration.
Lomas and Lipschultz (2006) initially hypothesized that rapid changes in the relative availability of light and nitrate in the surface ocean causes transition periods where double-peaked PNM temporarily exist. One peak would be the newly forming peak, and one would be degrading. Water column characteristics such as the distance between nitrite maximum and top of nitracline, or the concentration of nitrate at the nitrite maximum, may hint at whether the nitrite accumulation is newly forming or a remnant feature, but the age of the PNM remains difficult to discern using only this information. It is also difficult to tell if there is an age difference between the main peak and the smaller peak, and our rate measurement data is not high enough resolution to capture the smaller peak adequately. We are hopeful that our future

investigations of nitrite age near the PNM using natural abundance isotopes will be able to provide more insight.

**Comment:** The chlorophyll correlations in Figure 2a,b appear to be strongly influenced by a single extremely high data point. I wonder how the interpretation would change if the data were fit without this one point; it looks as though the slope would be quite a bit higher while still (maybe) being significant. It would be appropriate to check this and discuss model sensitivity.

Yes, the Station 8 chlorophyll maximum was much larger than the other stations. The significance of the relationship declines when Station 8 is removed from the regression (see figure below comparing regressions with and without Station 8), and it is worth adding a note in the revised manuscript to discuss the sensitivity of the results to the inclusion of Station 8, which has a disproportionately large impact on the regression line. Therefore, care needs to be taken when interpreting the importance of chlorophyll concentrations on formation of nitrite maximum as suggested by the reviewer.

- Added [353-354] "Removing the outliers from the two chlorophyll regressions (Fig. 2a, 2b) did not improve the correlations (R2 = 0.06 and 0.09, respectively)."

[Figure]

**Comment:** The last paragraph of the Conclusion section brings up nitrous oxide formation, yet it is not mentioned anywhere else in the manuscript. Information in the conclusion should wrap up the findings of the paper, not introduce new ideas. If desired, the authors could add a "forward looking"/"future work" paragraph at the end of the discussion that briefly fleshes out the ideas presented in the conclusion in light of their own data set, but I do not think it should be in the conclusion because it is not actually a conclusion of the work presented in this study.

Thank you for your comment. We agree, the discussion of nitrous oxide would be more appropriate in the Discussion section. The conclusions will be refocused on data/results directly presented in this manuscript and discussion of implications for nitrous oxide production will be moved to the discussion.

- Moved to discussion [823-830]
  "With nitrite production in the PNM predominantly linked to ammonia oxidation, this has potential implications for production of nitrous oxide in the upper water column of the ETNP. The ETNP is known to be an important source for atmospheric nitrous oxide (Babbin et al., 2020; Tian et al., 2020), with high accumulations of nitrous oxide in the near surface (Kelly et al., 2021; Monreal et al., 2022). Nitrous oxide production in the near-surface maximum has been linked to a combination of hybrid production from AOA, and bacterial denitrification (Kelly et al., 2021; Monreal et al., 2022; Trimmer et al., 2016). Thus, conditions that favor enhanced ammonia oxidation could also promote enhanced nitrous oxide production and emissions, thereby forming a link between stimulation of high primary productivity and high rates of nitrous oxide production and emission. "

**EGU Editorial Office**

**Comment:** Please ensure that the colour schemes used in your maps and charts allow readers with colour vision deficiencies to correctly interpret your findings. Please check your figures, especially figure 6, 7 and 8, using the Coblis – Color Blindness Simulator (https://www.color-blindness.com/coblis-color-blindness-simulator/) and revise the colour schemes accordingly. Your manuscript includes blue colored text. Please notice that this will not be included in the final paper.

- Figure 6 ,7 and 8 have been re-colored for color-blind accessibility. Blue colored text has been corrected. Note Figure 6 has been moved into the Supplement as Figure S3.

[Figure]

[Figure]

---

## Author Response (AR2)

**Associate Editor Comment:** Reconsider after Minor Revisions

Thank you for your consideration of this revised manuscript.

**Reviewer 1:**
**Overview:**
I reviewed an earlier version of the manuscript, and I raised my concerns primarily on the following:

      1) There's some unclearness on describing the method used for rate calculation in the earlier version, particularly if the rates were derived by integrating the paired light-dark incubation was not specified.

      2) The impacts of tracer enrichment (200nM) on the measured rates and the main findings were not sufficiently discussed.

In this revision, the authors have made major revisions in responding/ addressing these issues, and have taken most of my specific suggestions. I appreciate those efforts and am satisfied with the improvement in clarifying the method and adding the associated discussion on their results and findings.

In light that the authors have addressed my major concerns and improved the manuscript in many aspects, I am glad to recommend publication of the manuscript after incorporating a few minor comments/ suggestions (see below).

Thank you for all your effort to improve this manuscript. We truly appreciate your time and your thoughts.

**Specific comments:**
Lines 178-179: I am surprised that the sensitivity of the OPA method (Holmes et al., 1999) can meet the trace $NH_4^+$ measurement in the present study (i.e., many samples <50 nM of $NH_4^+$ in Table S1). Could you clarify the detection limit and accuracy of your $NH_4^+$ measurement? Or any instrumental or experimental improvements have been made?

Thank you. Ammonium measurements made with the OPA method have a detection limit of 40nM, and this is clarified in the updated manuscript. Some of the ammonium measurements presented in Table S1 are much lower than this limit. The detection limit is also mentioned in the Table S1 caption.

- Line 140: Detection limit for this ammonia method was 30 nM

- Supplement line 7-8 "Some ammonium concentrations reported were below the 30nM detection limit for the quantitation method used."

Lines 311: Since the PN was measured using the EA-IRMS, usually, at least 10 µg of N is required for stable and accurate PN concentration and isotope analysis. Here only 300mL of samples was used for PN samples; it should be clarified that the if PN concentration is high enough to get sufficient PN for EA-IRMS analysis.

Yes, the water budget for this experimental design resulted in low volumes for analyzing particulate N uptake at the end of the time course. To work around this, the replicate bottles were combined before filtration to increase total N on each filter. This means we do not have replicate measurements for nitrite uptake rates. I have clarified the combined volumes in the manuscript.

- Line 217-218: "At the end of the incubation (24 hr), the remaining ~300 ml of water in each replicate bottles was combined in order to maximize the amount of nitrogen available for isotope analysis."

-
  Supplement line 10-13 "Replicate bottles for nitrite uptake measurements were combined in order to maximize particulate nitrogen content prior to isotope analysis, and therefore do not have an associated standard deviation. Nitrite uptake rates have been bolded when particulate nitrogen content was below 10 ug"

Line 387: See my comment on PN measurement above. Additional information on the detection limit and accuracy of the EA-IRMS will be helpful.

Thank you. See above.

Line 517: As the primary nitrite maximum has been defined as PNM, unless you are referring to another nitrite maximum, the 'nitrite maximum' (here and in many other places) can be described as PNM for short and consistent.

Yes, we agree with your comment, it is useful to have short and consistent abbreviations. For this manuscript, Primary Nitrite Maximum is defined as the entire nitrite accumulation. When referring to the feature in general, PNM is used. Specific characteristics of the PNM feature are used to describe the depth and "size" of the peak of the nitrite profile; these are depth of maximum nitrite (m) and concentration of the nitrite maximum (µm).

Lines 573: The authors described that the PNM isopycnal varied from 24.1 in the coastal stations to 24.3 in the offshores stations (e.g., line 1086). You probably can use a bar to indicate the range of PNM/ or two dashed lines with different colors to denote the isopycnal in the coastal and offshore stations.

The mean isopycnal for the PNM was not statistically different between coastal and offshore, so I have removed the reference to the separate isopycnals (line 1086). A mean coastal PNM isopycnal can be added to a version of Fig. 4 (see below), but the data depicted on the figure is not separated by coastal and offshore, and since these mean PNM isopycnals are not statistically different, this change to the figure may not provide additional insight. In Fig. 4 we can get a sense of the range in depths of the nitrite maxima from the rates colored in pink (rates collected close to the nitrite maxima peak), so we have elected not to alter the final figure.

[Figure]

Lines 1219-1223: I agree with the statement. Nevertheless, it should be noted that other potential sources (e.g., NO2- production from the labile DONs by the ammonia oxidizers); or the rate measurements under conditions closer to the in-situ concentrations to reduce/ correct the potential enhancement should provide more comprehensive information on the balance of NO2- budget; and thus are encouraged in future studies.

Thank you for your comment. Yes, we will note other potential sources/losses of nitrite.

- Line 863-864: Some of the discrepancy between rates and observed nitrite accumulation may also be attributable to potential enhancement of rates from tracer addition, or nitrite production from other sources not captured in our tracer experiments.

Lines 1445-1446: Reference needed?

Yes, this idea has been supported with a citation.

- Line 971-974: The correlation found in the MLR analysis between the chlorophyll-nitrate interaction term and the nitrite maxima supports the idea that higher nitrite accumulation requires the presence of higher levels of nitrate within the chlorophyll bloom (Anderson and Roels, 1981; Collos, 1998).